# Effect of adaptive cruise control on fuel consumption in real-world driving conditions

Ayman Moawad [1] ✉, Matthew Zebiak[2,3], Jihun Han[1], Dominik Karbowski[1], Yaozhong Zhang[1] & Aymeric Rousseau[1]

This paper presents a comprehensive analysis of the impact of adaptive cruise control on energy consumption in real-world driving conditions based on a natural experiment: a large-scale observational dataset of driving data from a diverse fleet of vehicles and drivers. The analysis is conducted at two different fidelity levels: (1) a macroscopic trip-level benefit estimate that compares trips with and without cruise control in a counterfactual way using statistical methods, and (2) a situation-based comparison achieved through the segmentation of trips into distinct driving situations such as acceleration, braking, cruising, and other maneuvers. The results of this research show that the effect of cruise control on energy consumption varies across different driving situations and levels of analysis. In a macroscopic trip-level analysis, cruise control engagement is associated with a slight increase in fuel consumption across the fleet. As revealed later by the situation-based analysis, this result can be attributed to the negative impact of cruise control on energy consumption in cruising mode, which is the most common driving situation. However, the situation-based comparison demonstrates that cruise control can provide fuel consumption benefits in situations involving acceleration and braking, particularly when a preceding vehicle is present. The study also emphasizes the importance of controlling for various factors that can influence both fuel consumption and the likelihood of cruise control engagement to properly evaluate its effects.

Adaptive cruise control (ACC) has emerged as a promising technology in the realm of advanced driver assistance systems (ADAS) and has the potential to improve driving safety, enhance driver comfort, and reduce energy consumption. ACC automatically adjusts the speed of a vehicle to maintain a safe distance from the vehicle ahead. Nowadays, nearly all major automotive manufacturers offer ACC on their new vehicle models, however the impact of this technology on vehicle energy consumption has sparked debate.

Despite the growing body of research on the topic, there is still a need for further investigation using real-world driving data to better understand the real-world energy impact of ACC. Most studies performed to date have relied on limited datasets or simulation environments, which may not capture the full range of driving conditions, vehicle types, and driver behaviors that affect ACC performance and energy consumption in real-world scenarios.

For many decades, automakers could only improve vehicle energy efficiency by attacking physical sources of energy loss in a vehicle - aerodynamic drag, tire rolling resistance, engine friction, inertial losses due to mass, etc. In the past 20–30 years, the proliferation of electronic control systems and electrically controlled actuators in engines,

[1]Vehicle and Mobility Simulation department at Argonne National Laboratory, 9700 S Cass Ave, Lemont, IL 60439, USA. [2]2050 Partners, 81 Coral Drive, Orinda, CA 94563, USA. [3]Argonne National Laboratory, Lemont, USA. ✉e-mail: amoawad@anl.gov

transmissions, electric drive motors, and batteries has made significant optimization and loss reduction within propulsion systems possible through better software and control strategies.

However, energy efficiency in the real world is determined as much or more by driver behavior as by a manufacturer's engineering decisions. Anecdotally, it is possible for two drivers (e.g., a parent and teenage child) to achieve vastly different performance levels in fuel economy, despite driving the exact same vehicle on similar routes. Thus, marginal improvements in baseline vehicle efficiency could be easily outweighed by the inefficient habits of a particular driver.

Until very recently, automakers have had virtually no control over how their vehicles are driven by end users, and thus, they have had limited control of their products' use-phase energy efficiency. Automated driving technologies present the first opportunity to date for automakers to exercise a greater degree of control over the use-phase emissions of their products. In fact, this opportunity is actively growing; recent usage rates show that the share of distance driven using automated systems has increased significantly as systems have become capable of operating without interruption in more types of road and traffic environments.

This opportunity is the central motivation for examining the spectrum of human driving habits, as well as the capabilities and typical behaviors of today's automated driving systems. In understanding the relative strengths and weaknesses of the two populations (humans and automated driving systems), we can better inform the development of next-generation automated driving systems to deliver vastly improved efficiency.

Initial studies investigating the impact of ACC systems on energy consumption and efficiency showed promising results.[1] and[2] explored early research advances in adaptive cruise control, shedding light on its potential energy-saving benefits. More recent research has primarily focused on simulation studies[3] and test-track experiments[4]. The energy impact of ACC and other automated driving technologies has typically been analyzed in free-flow or car-following modes, often using artificially constructed scenarios and relying on the questionable capabilities of common models to produce realistic vehicle dynamics and/or driving behavior[5]. For instance[6], conducted a microsimulation study with a scenario-based approach, offering insights into the impact of automated vehicles on highway network emissions.

In general, the results have been mixed, depending on factors such as the tools employed, the methodology, the underlying control mechanisms, and the implementation[7,8]. For example[9], demonstrated in a meta-analysis of ACC's environmental impacts that the outcomes were highly sensitive to time gap settings, and various ACC control strategies that influence the results have also been identified[10,11] emphasized the importance of critically reviewing model assumptions and their practical applicability. Efforts to compare results and draw general conclusions from existing literature are challenging due to the differences in terminology, assumptions, scenarios, and evaluation criteria across studies. Simulation-based results are heavily dependent on internal models and assumptions, often focusing on theoretical potential in ideal conditions rather than on practical impacts. Experimental studies are potentially able to produce more reliable conclusions, but require more resources and offer more limited scope for generalizations. Experiments can also be prone to behavioral bias, where the participants change their behavior because they are aware that they are participating in a study. On this point, this particular study is able to avoid this bias because the data was collected in the background as part of GM's normal course of business, and the drivers did not have awareness of any studies that would make use of the driving data.

Moreover[12], highlighted in a systematic review the existing knowledge gap regarding interactions between human-driven and non-connected automated vehicles. Accurately representing these interactions in traffic models is challenging and can affect the results when assessing energy impacts.

The use of real-world driving data to analyze ACC systems' effects on energy consumption has become more prevalent and sophisticated in recent years. However, the literature on this topic remains limited. Despite advances in on-board measuring and high-performance computing, acquiring comprehensive data remains challenging.[13] notes the growing prevalence of ACC systems in modern commercial vehicles but highlights the lack of information on their operation and impact on traffic dynamics. They propose a unified data structure for easier comparison across various tests, vehicles, and systems. The complete dataset is published as an open-access database called OpenACC, which is planned to evolve as more tests are conducted. This project is at attempt to engage the scientific community in understanding ACC vehicles' properties and potential impacts on traffic flow and energy consumption, identifying key differences between ACC systems and human drivers, and helping design new ACC car-following models for traffic microsimulation.

Specifically, studies like[14] investigate the energy impact of ACC in real-world highway scenarios by comparing ACC driving behavior to human drivers. The research discovered that ACC followers contributed to string instability and had tractive energy consumption 2.7% −20.5% higher than human drivers individually and 11.2% −17.3% higher on a platoon level.[4] also examines the impact of ACC systems on traffic flow, energy consumption, and safety in real-world driving conditions through the testing of 10 ACC-equipped vehicles from different brands and powertrains at low speeds in various configurations. This study confirmed previous findings regarding the string instability of ACC systems, suggesting that their current form may lead to increased energy consumption. However, other researchers such as[15] found in a field test data evaluation that the fuel consumption rate for vehicles in ACC mode was about 5%–7% lower than for vehicles in non-ACC mode when traveling in similar conditions.

Despite the scarcity of literature and lack of consensus on the impact of ACC systems on energy consumption, particularly those utilizing real-world driving data, it is clear that the energy-saving potential of ACC systems can vary depending on factors such as traffic conditions, specific algorithms, driving conditions, vehicle type, and driver behavior. Further research is necessary to fully understand the potential energy savings and drawbacks of ACC in different driving scenarios, particularly on a larger scale and at a fleet level.

In this study, we extend the existing body of research by analyzing a large and diverse dataset of real-world driving data collected from a fleet of General Motors (GM) vehicles and drivers in the United States. Our dataset includes powertrain data, sensor and ADAS data, and GPS data at 1-Hz resolution, providing a rich and detailed account of vehicle performance, driving conditions, and ACC usage. This time-series data is augmented with (1) encrypted driver logs in order to uniquely identify drivers, (2) decoders to extract detailed vehicle information from VINs, and (3) map matching capabilities via HERE Maps to retrieve the surrounding driving environment and route-level information. This large observational study allows us to gain valuable insights into the real-world energy impact of ACC across a wide range of scenarios. Understanding the impact of ACC on energy consumption on a large scale and in a real-world setting can inform the development of future vehicle technologies that further improve fuel efficiency and reduce emissions, create better automated driving controls, and allow for the study of trade-offs between safety and efficiency.

The remainder of this paper is organized as follows: First, we present the results of our data analysis, and discuss our findings at two different levels: a macroscopic, trip-level analysis in which the results show ACC's effect on energy use over the entirety of the fleet; and a more granular, situation-based analysis that segments trips for a higher-resolution understanding of ACC impact. We then present a discussion of the results, their implications, and their limitations, and

**Table 1 | Linear mixed effect model estimates of the effect of adaptive cruise control engagement and other covariates on fuel consumption at the trip level**

| | Dependent variable | |
| --- | --- | --- |
| | FuelCons | p-value |
| Inverse Avg Speed (km/h)$^{-1}$ | 4.069*** | <2e–16 |
| (veh_spd_meanl) | (0.114) | |
| ACC Engagement | 0.260*** | 0.000438 |
| (ACC_engaged_cat) | (0.068) | |
| Max Vehicle Speed | 0.051*** | <2e–16 |
| (veh_spd_max) | (0.001) | |
| Inverse Trip Distance (1/km) | 6.154*** | <2e–16 |
| (dist_covered_kml) | (0.094) | |
| Trip Acceleration Energy | 4.583*** | <2e–16 |
| (veh_accel_nrg) | (0.045) | |
| Trip Elevation Change | 0.006*** | <2e–16 |
| (elev_delta) | (0.0001) | |
| Avg. Ambient Temperature | 0.001 | 0.504531 |
| (amb_temp) | (0.001) | |
| Avg. Engine Temperature | – 0.040*** | <2e–16 |
| (eng_temp) | (0.001) | |
| Constant | 3.543*** | <2e–16 |
| | (0.205) | |
| Observations | 40,507 | |
| Log Likelihood | – 78,939.110 | |
| Akaike Inf. Crit. | 157,912.200 | |
| Bayesian Inf. Crit. | 158,058.600 | |

*Note:* 7D3*$p < 0.1$, **$p < 0.05$, ***$p < 0.01$.
The main numbers represent the estimated effects evaluated using t-tests based on Satterthwaite's method; numbers in parentheses represent the standard errors, and reported p-values are two-sided based. Included are also measures of model fit and quality, and significance levels are denoted by *$p < 0.1$ (90% confidence), **$p < 0.05$ (95% confidence), ***$p < 0.01$ (99% confidence).

suggest directions for future research. Before concluding, we detail the methods used in the study.

## Results

In this observational study, we investigated the treatment effect of engaging ACC on vehicle fuel consumption while controlling for potential confounding factors. The primary objective was to determine whether the engagement of ACC resulted in a significant difference in fuel consumption.

We examined the factors influencing vehicle fuel consumption in L/100 km using a linear mixed effect model, as a statistical modeling technique that allows for the analysis of hierarchical and clustered data. Model details can be found in the Methods section. An analysis of the results in Table 1 reveals several significant relationships between the fixed effects and fuel consumption.

First, we observed a strong positive relationship between inverse average vehicle speed and fuel consumption. Here we note that inverse average vehicle speed is transformed as 60 times the inverse of speed, so that units are in min/km. Specifically, a 1-unit increase in this variable was associated with an increase of 4.069 units in FC ($t = 35.619$). This can be interpreted as every additional min spent on a km (decrease in trip speed) increases the FC by roughly an additional 4 L/100 km. This finding confirms previous statements that lower average trip speeds contribute to increased FC.

Additionally, we find other significant and insightful associations between other covariates and fuel consumption. Specifically, we note that elevation change exhibits a small positive association with FC. An

increase in elevation increases FC by +0.6 L/100 km for every added 100 m over the trip.

Also, we found a significant negative association between engine temperature and FC. Every 10-degree increase in engine temperature is linked to a decrease of 0.4 L/100 km in FC ($t = -50.737$). This result implies that higher engine temperatures are associated with lower fuel consumption due to improved engine efficiency at optimal operating temperatures.

We noticed that ambient temperature is not a statistically significant factor ($t = 0.667$) for fuel consumption change. This result indicates that higher ambient temperatures could marginally contribute to increased FC. Ambient temperature has a secondary effect on fuel consumption when engine temperature is controlled, as the latter has a more direct impact on FC.

The inverse distance covered showed a substantial positive relationship to FC. For each additional unit of inverse distance covered, fuel consumption increased by 6.1544 units ($t = 65.713$). This finding highlights the intuitive fact that as the distance traveled increases, FC usually improves (shorter trips generally exhibit higher fuel consumption due to the impact of cold start penalty, for example).

Also, it appears that a 1-unit increase in maximum vehicle speed corresponded to a 0.0513-unit increase in FC ($t = 89.367$). Provided everything else remains constant, this result suggests that vehicles reaching higher maximum speeds during a trip may consume more fuel. Similarly, trip level acceleration energy was another significant predictor of fuel consumption. A 1-unit increase in vehicle acceleration energy was associated with a 4.5827-unit increase in FC ($t = 102.702$). This result emphasizes that vehicles with higher acceleration energy levels at trip level are likely to consume more fuel.

Finally, our analysis revealed a significant ($t = 3.793$) treatment effect of engaging adaptive cruise control on fuel consumption. After controlling for the other variables in the model, we found that when the adaptive cruise control was engaged, fuel consumption increased by 0.26 L/100 km compared to when it was not engaged ($t = 3.793$). This result indicates that at the fleet level, the use of adaptive cruise control may lead to a slight increase in FC. All else being equal, ACC engagement has a negative impact on fuel consumption (on average, i.e., across all vehicles, drivers, speeds, etc.), with an FC increase of 0.26 L/100 km. We observed that the average FC across the fleet was 14.7 L/100 km; by tying back this number we can conclude from this result that ACC may present about 2% FC penalty on the fleet.

### Interaction terms

In this section we focus on including an interaction term between ACC usage and average trip speed. Although we observed a +0.26 L/100 km penalty on average across all trips, we can further investigate the ACC effect on fuel consumption as a function of trip speed to better understand the results. In that case, the ATE of ACC on FC we are trying to extract depends on and may vary with the different trip speed profiles:

$$\tau_i = Y_i(1) - Y_i(0)$$
$$= \beta_2(\text{ACC\_engaged\_cat}_{\text{TRUE}}) \qquad (1)$$
$$+ \beta_9(\text{ACC\_engaged\_cat}_{\text{TRUE}})(\text{veh\_spd\_meanl})$$

where $\tau_i$ represents the average treatment effect of ACC on fuel consumption for the $i$th trip, $Y_i(1)$ represents the fuel consumption with ACC engaged, and $Y_i(0)$ represents the fuel consumption without ACC engaged. $\beta_2$ captures the treatment effect of engaging adaptive cruise control, and $\beta_9$ is the introduction of an additional coefficient for the cross interaction term. The $\beta$s are new estimates from the results of a fitted model that includes interactions, with $\beta_2 = 4.96$ ($t = 14.96$) and $\beta_9 = -3.91$ ($t = -10.71$). Solving for a negative treatment effect on trip

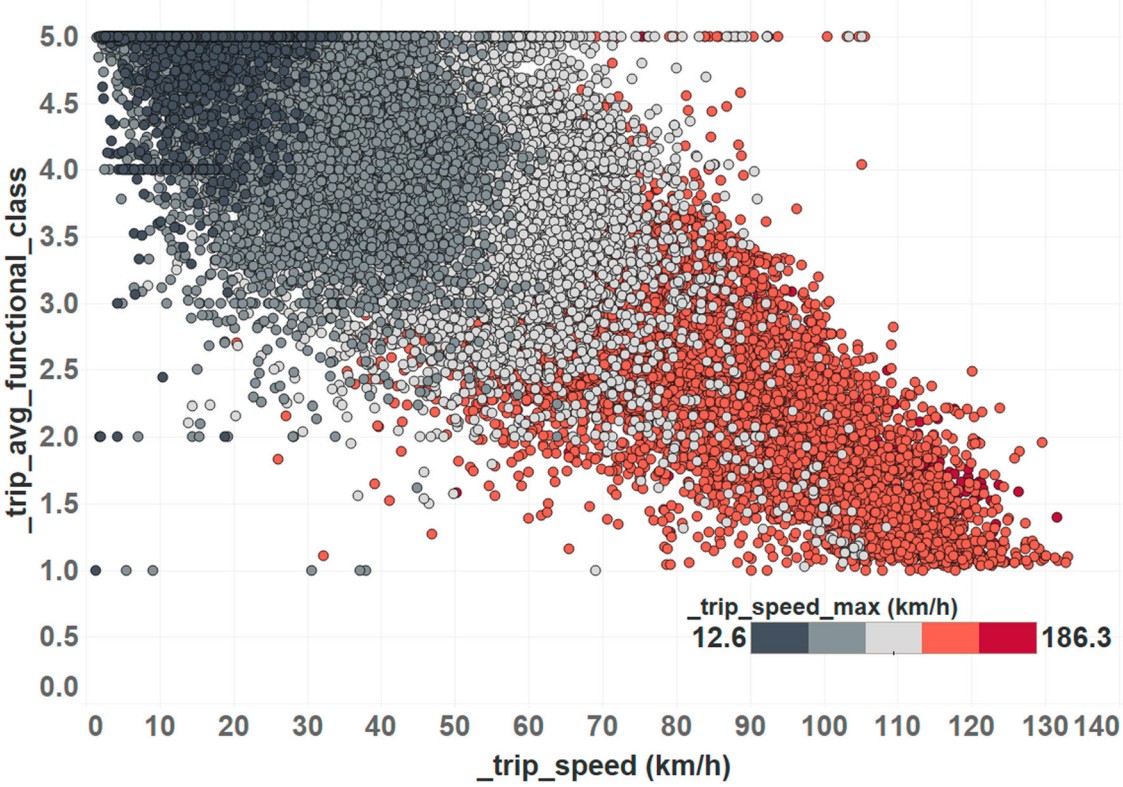

**Fig. 1** | Relationship between trip speed and trip topology by looking at average trip functional class structure.

mean vehicle speed $\bar{v}$ when ACC is engaged leads to the following:

$$\beta_2 + \beta_9 \times 1/\bar{v} < 0 \;\Rightarrow\; \bar{v} < 0.79 \qquad (2)$$

In this case, we find that the ATE of ACC on FC depends on a trip speed threshold of 0.79 km/min → ≈ 50 km/h; trips with lower average speeds see fuel consumption benefits from engaging ACC.

Further analysis (see Fig. 1) revealed that trips that average less than 50 km/h represent trips with the following characteristics:

- Higher functional class trips ( > 2.5 means trips that are mainly local/non-highway). Functional class is a road type indicator, reflecting traffic speed and volume, as well as the importance and connectivity of the road.
- Maximum speed < 90–100 km/h.

Our analysis here reveals that the effect of ACC on fuel consumption varies with average trip speed. While ACC engagement generally results in a slight increase in fuel consumption (+0.26 L/100 km), it tends to be more fuel-efficient at lower speeds, particularly below 50 km/h. This indicates that ACC systems can provide fuel consumption benefits in urban and suburban driving conditions. However, at higher speeds, the rigid speed maintenance of ACC leads to increased fuel consumption compared to human drivers. This interaction between ACC engagement and trip speed underscores the importance of considering different driving conditions when evaluating the energy impact of ACC systems. Additionally, these benefits are limited to a smaller number of trip profiles, which connects with the overall negative impact that was noted previously.

**Situation-level analysis**
The macro-level evidence presented above demonstrates that real-world use of ACC is overall not beneficial for trip-level fuel consumption. Furthermore, when benefits are present, they appear to be limited

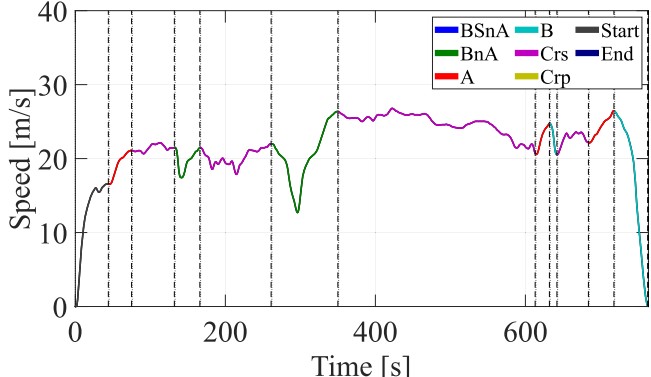

**Fig. 2** | **Example of trip situation segmentation using the standard U.S. Environmental Protection Agency highway drive cycle.** (Crs = Cruise, BSnA = Brake, Stop & Accelerate, BnA = Brake & Accelerate, A = Accelerate, B = Brake, Crp = Creep).

to certain trip conditions. A lower level analysis is needed to reinforce these findings and provide more granular explanations for ACC's impact on FC in distinct road and traffic conditions.

On a given trip, vehicle speed changes tend to be caused by certain external factors, under specific situations due to road events. Examples of these situations include braking, stopping, and accelerating due to red lights or stop signs, cruising for a while, braking and accelerating due to red lights or sudden lane changes by a preceding vehicle, and so on. Isolating those events by segmenting a given trip into specific situations allows us to obtain more targeted ACC benefit estimates for specific maneuvers, improving our overall understanding of the system and providing a more nuanced analysis.

Various situations occur in the driving of each trip. Figure 2 shows an example of a composite drive cycle (in this case, the EPA HWFE

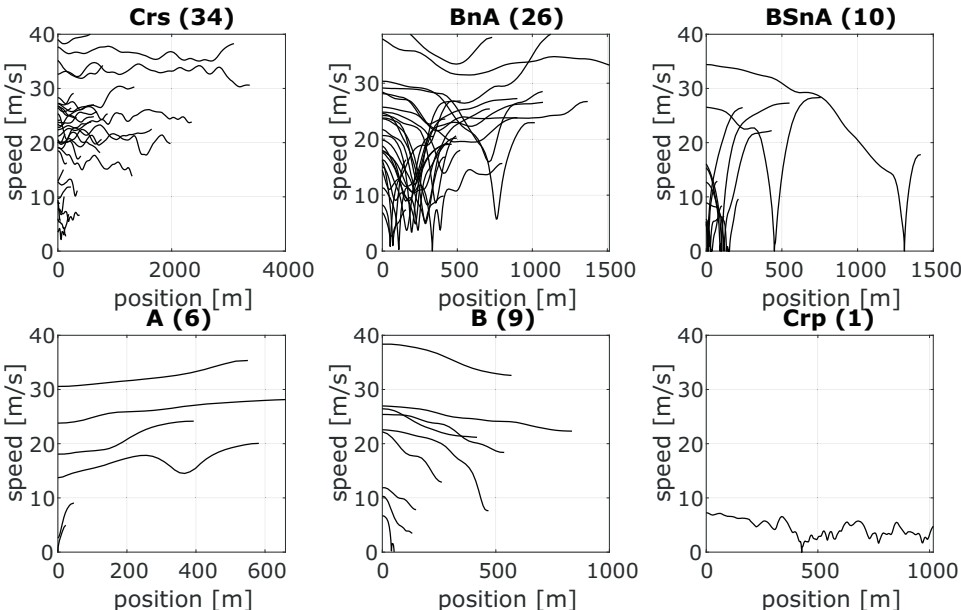

**Fig. 3 | Exemplary figure of speed trajectories corresponding to each maneuver obtained after situation segmentation on a given trip.** (Crs = Cruise, BSnA = Brake, Stop & Accelerate, BnA = Brake & Accelerate, A = Accelerate, B = Brake, Crp = Creep).

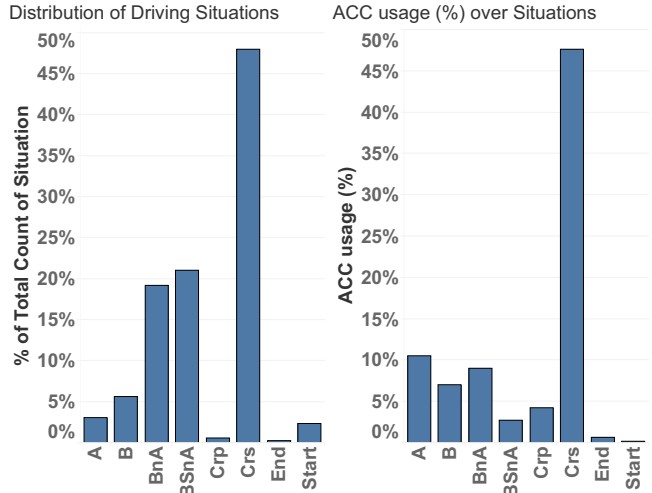

**Fig. 4 |** Percentage of driving over each situation, i.e., share of each situation over entire dataset (left), distribution of adaptive cruise control usage over situations (right).

cycle), after it has been run through our situation segmentation algorithm and broken up into distinct maneuvers. More details on the algorithm can be found in[16].

The algorithm enables situation-level and driving-level processing of the trips for the purpose of trip segmentation. By leveraging signals such as time, speed, yaw rate, position, acceleration, brake pedal position and accelerator pedal position, relative distance and speed with respect to the preceding vehicle, as well as ACC status, modes and settings, we can detect distinct situations within trips. We identified six different important situations, which we define as follows:

- Cruise (Crs): Maintain speed with little variation.
- Brake and Accelerate (BnA):

    – Brake and accelerate again without stopping.
    – Due to traffic lights, turns, roundabouts, etc.
- Brake, Stop, and Accelerate (BSnA):

    – Brake, stop completely, and accelerate again.
    – Due to traffic lights, stop signs, etc.
- Acceleration (A): Accelerate due to speed limit increase.
- Braking (B): Brake due to speed limit decrease.
- Creeping (Crp): Move forward at very low speed with some stops.

Figure 3 provides an illustration of the situations detected over a given trip. For added nuance, situations involving braking can be further split into brake events with and without turning, and, more importantly, situations in which the driver is aware of the preceding vehicle's status. In fact, it is also relevant to separate situations with and without the presence of a preceding vehicle.

### Situation-level results

In Fig. 4 we show the distribution of detected driving situations over trips, as well as the distribution of ACC usage over these maneuvers. The figure shows that the most common driving situation is cruising mode, accounting for 50% of driving time. BnA and BSnA are observed with almost equal frequency in the dataset. The creeping situation is seldom detected by the algorithm. We also note that, as expected, ACC is predominantly used in cruising mode.

It is important to note that situation segmentation enhances the resolution of our analysis and yields a larger number of data points. Consequently, we observe a significant increase in the number of situations generated during a trip. This increased sample size ultimately leads to better model coefficient estimates, asymptotic statistical efficiency, and consistency.

For our analysis, we employ a linear mixed-effect model similar in structure to the one used in the macroscopic study, with some modifications to the variable selection design. In the earlier trip-level analysis, we controlled for acceleration energy to normalize the trip. This was acceptable at a macroscopic level, but with shorter and more stable segments, we need to ensure that we do not double count the effect of aggressiveness on FC in relation to ACC. To do this, we introduce four new variables to make situation segments more directly comparable: average speed, entry and exit speeds over the segment, and minimum and maximum speeds during the segment. We also account for variability in thermal conditions, such as engine and

**Table 2 | Linear mixed effect model estimates of the effect of adaptive cruise control engagement and other covariates on fuel consumption across six predefined situations**

| | Dependent variable: Fuel Consumption | | | | | |
|---|---|---|---|---|---|---|
| | (1) Cruise | (2) Brake, Stop & Accelerate | (3) Brake & Accelerate | (4) Accelerate | (5) Brake | (6) Creep |
| Inverse Avg Speed (km/h)$^{-1}$ | 5.594*** | 2.956*** | 2.035*** | 4.082*** | 3.406*** | 2.931*** |
| | (0.152) | (0.121) | (0.190) | (0.216) | (0.125) | (0.127) |
| | [<2e−16] | [<2e−16] | [9.27e−15] | [<2e−16] | [<2e−16] | [<2e−16] |
| ACC Engagement | 0.142*** | − 0.316*** | −0.278*** | −0.714*** | 0.334*** | −1.175 |
| | (0.017) | (0.060) | (0.026) | (0.089) | (0.037) | (1.599) |
| | [<2e−16] | [1.21e−07] | [<2e−16] | [8.28e−16] | [<2e−16] | [0.4629] |
| Inverse Trip Distance (1/km) | 0.314*** | 2.880*** | 2.459*** | 2.777*** | − 0.179*** | 0.460** |
| | (0.003) | (0.014) | (0.007) | (0.018) | (0.008) | (0.189) |
| | [<2e−16] | [<2e−16] | [<2e−16] | [<2e−16] | [<2e−16] | [0.0152] |
| Starting Vehicle Speed (km/h) | −0.129*** | −0.137*** | −0.117*** | 0.533*** | −1.401*** | −0.161** |
| | (0.001) | (0.001) | (0.001) | (0.040) | (0.052) | (0.065) |
| | [<2e−16] | [<2e−16] | [<2e−16] | [<2e−16] | [<2e−16] | [0.0136] |
| Ending Vehicle Speed (km/h) | 0.151*** | 0.248*** | 0.234*** | − 1.519*** | 0.588*** | 0.081 |
| | (0.001) | (0.001) | (0.001) | (0.084) | (0.015) | (0.062) |
| | [<2e−16] | [<2e−16] | [<2e−16] | [<2e−16] | [<2e−16] | [0.1898] |
| Min Vehicle Speed (km/h) | −0.059*** | 0.072 | −0.114*** | −0.739*** | −0.567*** | |
| | (0.002) | (0.274) | (0.0005) | (0.040) | (0.015) | |
| | [<2e−16] | [0.7916] | [<2e−16] | [<2e−16] | [<2e−16] | |
| Max Vehicle Speed (km/h) | 0.127*** | 0.125*** | 0.084*** | 1.891*** | 1.381*** | 0.007 |
| | (0.002) | (0.002) | (0.001) | (0.083) | (0.052) | (0.110) |
| | [<2e−16] | [<2e−16] | [<2e−16] | [<2e−16] | [<2e−16] | [0.9463] |
| Trip Elevation Change | 0.037*** | 0.300*** | 0.240*** | 0.370*** | 0.264*** | 0.196** |
| | (0.0003) | (0.003) | (0.001) | (0.004) | (0.002) | (0.085) |
| | [<2e−16] | [<2e−16] | [<2e−16] | [<2e−16] | [<2e−16] | [0.0212] |
| Avg. Ambient Temperature | − 0.009*** | 0.093*** | 0.022*** | 0.036*** | 0.009*** | 0.333*** |
| | (0.001) | (0.002) | (0.001) | (0.003) | (0.001) | (0.039) |
| | [<2e−16] | [<2e−16] | [<2e−16] | [<2e−16] | [1.14e−09] | [<2e−16] |
| Avg. Engine Temperature | − 0.050*** | − 0.123*** | − 0.084*** | − 0.108*** | − 0.028*** | − 0.353*** |
| | (0.0005) | (0.001) | (0.001) | (0.001) | (0.001) | (0.036) |
| | [<2e−16] | [<2e−16] | [<2e−16] | [<2e−16] | [<2e−16] | [<2e−16] |
| Constant | 0.294* | −0.890** | 2.265*** | −5.405*** | 4.575*** | 37.389*** |
| | (0.162) | (0.348) | (0.218) | (0.440) | (0.162) | (4.355) |
| | [0.0735] | [0.0118] | [<2e−16] | [<2e−16] | [<2e−16] | [<2e−16] |
| Observations | 283,589 | 126,871 | 249,879 | 38,730 | 79,775 | 961 |
| Log Likelihood | −710,619.700 | −406,352.400 | −698,783.900 | −118,174.300 | −210,477.500 | −3,847.556 |
| Akaike Inf. Crit. | 1,421,271.000 | 812,736.700 | 1,397,600.000 | 236,380.600 | 420,986.900 | 7725.111 |
| Bayesian Inf. Crit. | 1,421,440.000 | 812,892.800 | 1,397,767.000 | 236,517.600 | 421,135.500 | 7798.131 |

*Note:* 7D3 *p < 0.1, **p < 0.05, ***p < 0.01.
The main numbers in bold represent the estimated effects evaluated using t-tests based on Satterthwaite's method; numbers in parentheses represent the standard errors, and included in brackets are two-sided based p-values.

ambient temperature, as well as changes in elevation and segment distance.

Table 2 reveals several interesting findings (due to the small sample size, the Crp situation data were deemed unreliable and have been excluded from the analysis):

– As elevation increases over a segment, FC also increases. This effect is most pronounced in acceleration situations and less so in BSnA and BnA situations. In cruising mode, a change in elevation has one-tenth the effect. Specifically, for every meter of change in elevation, FC is penalized by +0.37 L/100 km if the vehicle is in strong acceleration mode but only +0.037 L/100 km when cruising.

– Higher engine temperatures primarily benefit BSnA modes, with a decrease in FC of −0.12 L/100 km. In BnA situations, the benefit is slightly lower at −0.084 L/100 km. This is likely due to the absence of idling events (no stop), since engine temperature is a dominant factor in idle fuel rates. In cruising mode, the impact is smaller, with a decrease of only −0.05 L/100 km.

– The effect of ACC on fuel consumption varies depending on the situation. In cruising segments, the engagement of ACC results in a slight increase in FC (+0.14 L/100 km). In braking situations, the penalty that ACC offers is less clear (+0.334 L/100 km); however, we hypothesize that human drivers are better able to leverage coasting before an actual brake event, which may lead to efficiency benefits

as the nominal fuel consumption of a deceleration event is spread over a greater distance traveled. Furthermore, some human drivers might utilize multi-anticipation, reacting to more than one vehicle ahead[17].

– Engaging ACC for acceleration-involved situations (such as BSnA, BnA, and A) seems to provide advantageous FC benefits. This is because the positive impact of ACC on FC during pure acceleration outweighs the negative impact observed during braking.

Supplementary Table 1 presents results with an additional layer of detail that differentiates maneuvers based on the presence or absence of a preceding vehicle. In all situations where no preceding vehicle is present during the segment, engaging ACC appears to increase FC. Conversely, when a vehicle is present, the engagement of ACC can provide some benefits, with the exception of braking situations. This can be calculated by combining `ACC_engaged_cat` and `veh_ahead_cat` along with their interaction term coefficient. The negative interaction term in all situations (except braking) suggests that ACC is advantageous when engaged in the presence of a preceding vehicle. It is important to note that the number of data points is significantly reduced in this design, leading to marginally statistically significant estimates in some cases.

## Discussion

This research has investigated the impact of adaptive cruise control on fuel consumption, shedding light on how this technology can affect driving efficiency. Our findings contribute to the growing body of literature on the subject, which includes several studies that have analyzed the impacts of ACC and other advanced driver assistance systems on fuel efficiency and emissions.

In examining the segmented results in Supplementary Table 1, we find that certain maneuvers (BnA, BSnA, A) are better executed by automated driving systems, while others (Crs, B) are better executed by humans. The automated driving system's efficiency is also dramatically affected by the presence or absence of another vehicle ahead. In focusing on the dynamics of these individual maneuvers, we can assess the underlying causes of the efficiency discrepancies and theoretically design an automated system that can outperform humans in all types of driving maneuvers and road conditions.

### Open-road cruising

According to Table 2, we observe a 0.14 L/100 km fuel consumption penalty for engaging adaptive cruise control in cruising situations. At first, the fact that current cruise control systems are less efficient on average than human drivers in "cruising" maneuvers may seem counter-intuitive. However, closer inspection of a typical cruising scenario illuminates the reasons for the relative shortfall of automated driving systems and presents opportunities for future improvement.

First[18], asserts that "in today's cruise control systems, substantial energy is wasted by rigidly controlling to a single set speed regardless of the terrain or road conditions, [and] significant improvements in fuel economy and EV range can be achieved by relaxing the requirement that cruise control maintain a single constant speed at all times." We hypothesize that human drivers benefit from more flexibility compared to automated systems while in a cruising mode—that is, they tend to hold relatively constant pedal position and allow vehicle speed to vary slightly (often without noticing), particularly over changing terrain and in open-road conditions where traffic is not a significant concern. This allows for more steady-state operation, which is particularly advantageous in ICE applications that can experience step changes in efficiency (e.g., powertrain downshifts, engine operating mode changes) as a result of small changes in vehicle load.

The mechanisms for improving efficiency through flexibility in cruise control were explored in detail in ref. 18. In this experimental study, a modified cruise control system was designed to let its speed vary within defined limits ( ± 8 km/h) in response to changing road grades. This modified cruise control was tested back-to-back against standard cruise control on a grade schedule (taken from US-23 in Michigan) programmed into a dynamometer at the GM Proving Grounds. The study found that the modified cruise control uniformly achieved higher fuel economy than standard cruise control on all tested vehicles, by an average of 3.5% for the gasoline vehicle, 3.9% for the diesel vehicle, and 3.8% for the electric vehicle. It achieved these gains primarily by limiting engine braking on declines, limiting powertrain downshifts on inclines, and reducing overall tractive power requirements on inclines by around 15% by capping engine torque increases and allowing vehicle speed to drop temporarily. As automated driving systems evolve from simplistic cruise control to Level 2/3+ autonomy, there is evidence that human occupants are more tolerant of the system changing the cruising speed without human input. Therefore, future automated driving systems should fully capitalize on this flexibility that is deemed acceptable by passengers to achieve gains in energy efficiency.

We plan to conduct further investigations to support and explain the underlying mechanisms. Specifically, we are undertaking two studies: one analyzing the energy-saving benefits of ACC against different driver profiles, and another leveraging machine learning methods to model the relationship of vehicle dynamics to energy consumption with and without ACC at a microscopic level (second-by-second analysis). These studies will enhance our understanding of the efficiency improvements and provide more evidence for the hypotheses discussed in this paper.

### Dynamic maneuvers and driving with vehicles ahead

Supplementary Table 1 expands on the results of Table 2 by including the presence of a vehicle ahead as an interaction term. Introducing this new factor results in some notable changes in the effects of ACC engagement on fuel consumption. We now see fuel consumption penalties for engaging adaptive cruise in all studied maneuvers (no vehicle ahead), whereas Table 2 showed some benefits for more dynamic maneuvers such as BnA and BSnA. However, in examining the ACC engaged effect only in cases where there is a vehicle ahead (by accounting for the interaction term), we observe fuel consumption benefits, though some maneuvers do not have statistically significant main effects (e.g., ACC system braking event with no vehicle ahead is not part of the technology). In other words, while ACC is less efficient than humans on average in the examined dataset, it is more efficient than humans on average when it is following another vehicle. This is a significant finding, and one that refines our understanding of the mechanisms of energy savings in automated driving. Our hypotheses regarding the differences in ACC impact on FC in open-road vs. following conditions are as follows.

### Cruising with vehicles ahead

In cruising maneuvers, open-road ACC suffers efficiency penalties (+0.14 L/100 km) as a result of its rigid control to a single set speed at all times. However, in the presence of a vehicle ahead, ACC allows vehicle speed to drop below the driver's set speed in order to maintain a comfortable following distance to the vehicle ahead. Therefore, in cases where human-driven vehicles ahead may naturally slow down due to inclines or other external factors, a vehicle with ACC will correspondingly slow down to maintain an appropriate following distance. In effect, the system temporarily mimics the more efficient behavior of the human driver ahead, and claims the associated efficiency benefits. On the other hand, a vehicle with ACC does not experience a symmetric FC penalty in cases where a human driver ahead is less efficient than ACC (e.g., surpassing the driver's set speed) since it is not permitted to exceed its set speed.

We assert that this asymmetric opportunity for efficiency improvement is the core driver of ACC's energy savings while cruising

in the presence of a vehicle ahead—the system matches the behavior of efficient human drivers ahead and collects the associated savings, but does not match the behavior of less efficient drivers ahead and thus avoids the associated penalties.

## Dynamic maneuvers (BnA, BSnA)

In BnA/BSnA maneuvers, which are common in dense traffic (including stop-and-go scenarios), we assert that there is a similarly asymmetric opportunity to improve efficiency through both flexible speeds and flexible acceleration rates. Open-road ACC has predefined calibrations that determine how the vehicle accelerates/decelerates in response to changes in set speed or initial engagement of ACC. Typically, these calibrations are set fairly aggressively, so that the vehicle achieves the driver's requested set speed as quickly as possible. Even with these fairly aggressive calibrations, ACC engagement results in an average −0.3 L/100 km impact to fuel consumption across the entire dataset. When there is a vehicle ahead, ACC is able to reduce fuel consumption even further in BnA and BSnA maneuvers. This is because ACC cannot ever accelerate any more aggressively than the open-road calibration limit even when a human driver ahead is particularly aggressive, but it has the opportunity to accelerate much more efficiently when following an efficient driver. This asymmetry results in a net savings in these maneuvers, when automated driving systems follow a sufficiently large number of distinct drivers with different behaviors.

## Strict acceleration maneuvers

Table 2 and Supplementary Table 1 show that ACC engagement during strict acceleration maneuvers leads to a reduction in fuel consumption of −0.71 L/100 km. We contend that these savings arise from a calibrated limit of allowed acceleration while ACC is engaged. While human drivers are able to command up to the full capability of the engine during an acceleration maneuver (and incur massive fuel consumption penalties for doing so), ACC is limited to a maximum acceleration value that is much lower than the vehicle's full capability, even in open-road conditions. The data supports this point − the maximum acceleration observed in the dataset is 7.5 m/s$^2$ for ACC, just about half of the 14.7 m/s$^2$ maximum for human drivers. Likewise, the median positive commanded acceleration is 0.187 m/s$^2$ for ACC compared to 0.316 m/s$^2$ for human drivers (41% lower while in ACC). If we focus only on events where ACC is following a vehicle ahead, FC savings in acceleration maneuvers are significantly greater. This is another result of the asymmetric upside potential discussed in the previous section; ACC benefits from following efficient drivers, but incurs no penalty relative to open-road ACC for following inefficient drivers.

## Strict braking maneuvers

Table 2 and Supplementary Table 1 show that ACC engagement during strict braking maneuvers leads to an increase in fuel consumption, +0.33 L/100 km. In these maneuvers, we hypothesize that this FC penalty is largely a result of ACC's high deceleration rates. These rates are intentionally set high by manufacturers because the downside risks of insufficient deceleration in cruise are severe. However, there is some opportunity to make these default deceleration rates less conservative (and thereby, more efficient) in future systems as sensing capabilities and control systems improve.

In open-road conditions, deceleration events can only be triggered by a decrease in the driver's requested set speed. As mentioned in an earlier section, the deceleration rates commanded in these maneuvers are predefined in calibration tables, and are generally set to be aggressive so the vehicle quickly responds to the driver's command. This means that braking maneuvers in ACC are generally shorter in both time and distance compared to equivalent maneuvers executed by human drivers. We see this reflected in the median deceleration commanded during braking events, which is −0.2 m/s$^2$ for ACC and −0.18 m/s$^2$ for human drivers.

In cases where there is a vehicle ahead, deceleration events are mostly triggered by the vehicle ahead slowing down. We observe from Table 2 and Supplementary Table 1 that the penalty for braking events when there is a vehicle ahead is about 30% less than the dataset average. We hypothesize that this is a result of the reduced capacity for a driver to coast when there is a slowing vehicle ahead. In other words, human drivers tend to slow down more rapidly when there is slowing traffic than in open-road conditions, so the capacity for human drivers to coast and outperform ACC shrinks in these particular situations but, notably, is not eliminated entirely.

## Limitations of the study

Our study in its current form has some limitations.

**Causality.** As an observational study, it cannot establish causality. While we have attempted to control for various factors, it remains possible that unobserved variables may have influenced the results. We feel that the hypotheses presented in the subsections above are the most plausible explanations for the observed impacts of ACC on fuel consumption in certain maneuvers, but further, more direct A-B comparisons of humans and automated driving systems in these specific maneuvers would be required to definitively establish the root causes for the observed phenomena.

**Macroscopic vs microscopic analysis.** There is a need for high-resolution, microscopic-level analysis (i.e., second-by-second) to better understand the nuances of ACC's impact on fuel consumption. Future research should explore these aspects in greater detail to validate and extend our findings. Specifically, more granular analyses of traffic conditions, powertrain types, differences in ACC settings or potentially control types, and regional regulatory differences in ACC performance are needed to build a more comprehensive understanding of the factors influencing fuel consumption in ACC.

The findings from the macroscopic trip level and the situation-based findings are interconnected and reinforce each other. While the macroscopic analysis indicates a slight increase in fuel consumption across the fleet when ACC is engaged, the situation-based results reveal that ACC has a negative impact on energy consumption specifically in cruising. Given that cruising is the most prevalent driving situation, the fuel penalty observed in the macroscopic analysis can be attributed primarily to the increased fuel consumption during cruising with ACC. This connection between the two levels of analysis highlights the importance of examining the effects of ACC on energy consumption in different driving situations to gain a comprehensive understanding of its overall impact.

**Data representativeness.** The representativeness of our sample is also a potential limitation, as it consists of a single fleet of vehicles primarily used by GM employees and engineers. Although our data covers a large area of the U.S., it is not guaranteed that the findings can be generalized to the broader population. We believe that we have presented, in the data section, the details, key statistics and distributions pertaining to the population of vehicles in this study for full transparency.

**Data quality.** The accuracy of our results is contingent upon the quality of the sensors, data collection processes, and the algorithms involved in the analysis. Despite our diligent data processing and cleaning, these factors may have introduced some degree of undetected error or bias.

**Data sufficiency.** The sufficiency of data for the ATE analysis is an important aspect, especially for researchers that would be interested in conducting such a study in the future. We did not conduct a specific data sufficiency study to determine the minimum amount of data

needed for consistent ATE results. However, we can provide some general insights based on statistical principles.

More data generally leads to better results in statistical analysis. As the sample size (N) increases, the standard error decreases, leading to more precise estimates and reduced uncertainty. This principle, known as statistical power, is crucial for detecting small effects, such as the impact of ACC on fuel consumption. When the effect signal in the data is small, a significant amount of data is needed to overcome the noise. Additionally, statistical consistency implies that as the sample size increases, the bias in the estimations is reduced, providing more reliable results.

The effect of ACC on fuel consumption is relatively small, necessitating a significant amount of data to detect the signal amid the noise. But also, we control for many variables in our analysis, effectively slicing the data space across multiple dimensions. To ensure statistical significance and meaningful results in this high-dimensional space, a large number of data points are needed. Without sufficient data, the hypercubes within this multidimensional space would lack enough points to draw reliable conclusions.

It is worth noting that conducting a data sufficiency study is not as simple as it may seem, as there are various ways the data could be sampled, including random sampling, cluster sampling, or stratified sampling. Each of these methods can introduce different biases and complexities, requiring a sophisticated study design to offer relevant recommendations. In our case, we are dealing with a natural experiment from a purely observational study, making it challenging to predict how the results might differ under alternative sampling methods. Future research could include a well-crafted data sufficiency study to determine the minimum data requirements for robust ATE analysis. This study could include randomly reducing the existing dataset (e.g., by 20 percent) to test if the main findings are still uniformly observed in all reduced datasets.

### Future research directions

Our research offers valuable insights that can inform vehicle manufacturers, policymakers, and drivers about the potential effects of ACC on fuel consumption. Our findings indicate that ACC has the capacity to dramatically impact vehicle energy efficiency in both positive and negative directions, depending on driving situations, system behavior, and the presence of a vehicle ahead.

Much as flexibility in cruising speed can allow for more steady-state operation over changing grades, greater flexibility in following distance (instead of a single, driver-selectable "gap setting" as is common in today's vehicles) can allow for steadier, safer, more efficient operation in highly dynamic traffic conditions. For example, an automated driving system that detects a severe slowdown far ahead in its lane of travel could choose to coast and slow the vehicle pre-emptively, rather than travel at the set speed and command a severe braking event only when the vehicle is at imminent risk of not maintaining its minimum following distance.

Both human and automated drivers can theoretically pay attention to the trajectories of several vehicles ahead, scan multiple lanes of traffic, and modulate their speed proactively to prevent severe and costly braking and acceleration events. In practice, we find that few human drivers put in this level of thought and effort to achieve an efficient ride, but automated systems have the potential to consistently operate with high efficiency in traffic, if their sensing and planning capabilities are fully leveraged to achieve this objective. The tested ACC systems track both the first and second vehicle ahead as separate objects, and the position, velocity, and acceleration of these vehicles can be used as separate, distinct inputs in the ACC system's decision-making process. Particularly when traffic conditions are at their most unpredictable, automated systems can benefit substantially from their multi-modal sensing capabilities, which are always active and free of the distractions that can affect human drivers. When

powerful data about surrounding traffic conditions is combined with a proactive control system designed to limit unnecessary speeding and acceleration in traffic, automated driving systems have the potential to vastly outperform human drivers in terms of safety, comfort, and energy efficiency.

Overall, this study provides a deeper understanding of the interplay between ACC and fuel consumption in various driving situations. Our findings underscore the importance of considering the broader context when assessing the impact of advanced driver assistance systems. Future research should focus on overcoming the limitations of this study by conducting more controlled experiments, investigating a wider variety of vehicles and driving conditions, and refining data collection and analysis methods. Such research will contribute to the ongoing efforts to optimize ACC systems and other advanced driver assistance technologies for improved fuel efficiency and reduced greenhouse gas emissions.

## Methods

### Overview and comparison to prior work

The work presented in this paper uses a significant amount of real-world data in various environments rather than simulations or small-scale, real-world experiments described earlier. First, we will take a close look at the previous related work conducted by GM in 2019[19]. The approach, the data, and the methodology will be detailed to underscore how this current study extends the existing research.

In previous research conducted by GM engineers[19], data were collected from the 2018 Cadillac CT6, the first vehicle with Super Cruise technology, which combined ACC and advanced lane-keeping functionality using cameras, sensors, and GPS locators. The study involved 51 vehicles driven by employees on their daily commutes for 62 days between November 16, 2017, and January 16, 2018, covering 320,742 km in 13,416 trips. The data contained information on fuel consumption, vehicle speed, and ACC state collected at a 1-Hz rate.

In this previous study, the researchers analyzed the impact of ACC on energy consumption by comparing fuel values when ACC was ON versus OFF at various speed intervals across the entire fleet. The method involved aggregating fuel consumed per mile at each speed interval for vehicles with ACC engaged and those without ACC engaged, regardless of differences in vehicle models, drivers, or trip/ driving conditions. To account for differences in ACC usage and distance covered at various speeds, the researchers adjusted the fuel consumption benefits based on utilization rate and local distance traveled at each speed. The raw delta fuel consumption benefit was then weighted by the proportion of driving done at that speed interval relative to the total driving distance, resulting in a weighted adjusted average.

The method was deemed effective and the approach validated given the limited potential biases, the limited number of vehicle models and drivers involved in the study, and the extended period of data collection.

In our current study, we use a larger dataset collected from a fleet of 157 vehicles equipped with either traditional ACC or Super Cruise technology, noting that the longitudinal control system is identical between the two. Our extensive dataset includes 40,356 trips, covering 1,094,215 kilometers and 16,389 hours of driving by 95 different drivers. The data collection efforts are ongoing, but the results in this analysis cover the period of July 1, 2021, to September 1, 2022. With this richer dataset and larger fleet, we obtained more accurate "real-world" estimates of ACC benefits.

However, to accurately isolate the true effect of ACC on energy consumption and mitigate potential biases in our findings, we have relied on a statistical approach with carefully controlled variables. This refined method enhances our ability to discern and quantify the energy-saving potential of ACC technology across a variety of real-world driving conditions.

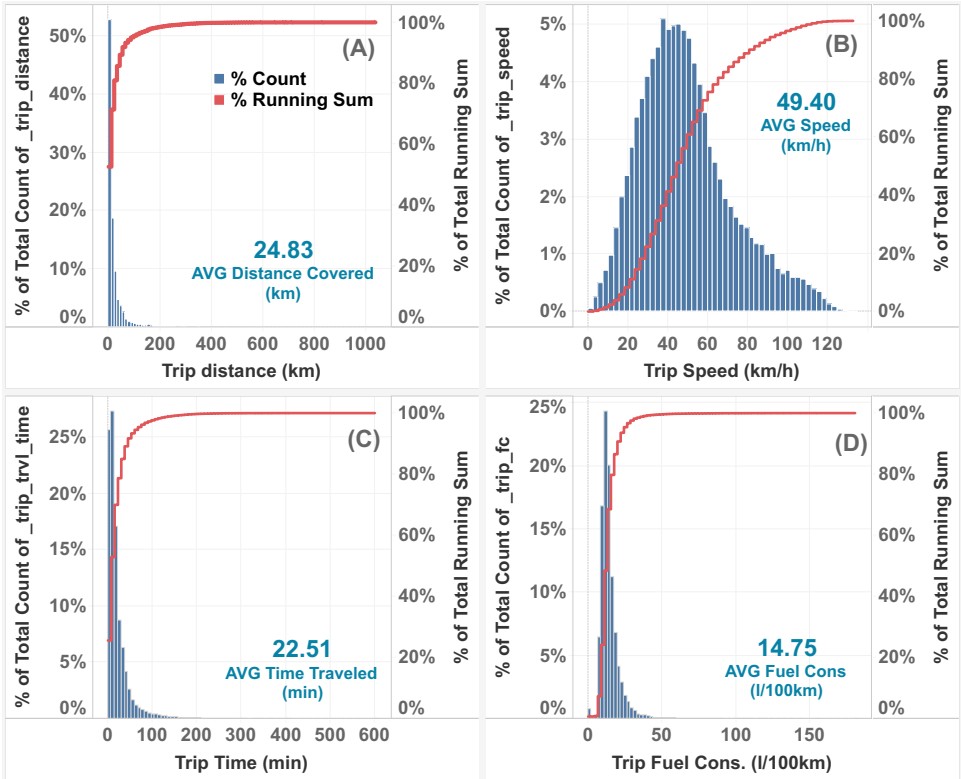

**Fig. 5 | Fleet-level distribution of key trip-level data. A** Trip Distances, (**B**) Average Trip Speeds, (**C**) Travel Times, and (**D**) Fuel Consumption. These histograms provide a comprehensive overview of the dataset, illustrating the range and distribution of the variables analyzed in this study. Trip Distances show most trips under 50 km with an average of 24.8 km, Average Trip Speeds peak around 125 km/h with an average of 49.4 km/h, Travel Times are mostly under 50 minutes with an average of 22.5 min, Fuel Consumption is mostly below 30 L/100 km with an average of 14.7 L/100 km but some extreme outlier trips.

## Data collection and management

In collaboration with GM through a Cooperative Research and Development Agreement (CRADA), we collected a large-scale dataset of real-world driving data, as described above. The dataset includes over 60 different signals at 1-Hz resolution, such as powertrain data (e.g., engine, fuel, transmission, thermal, etc.), automated driving assistance data (e.g., ACC, lane-keeping, gap settings, etc.), sensor data (e.g., relative lon/lat distance/speed with the vehicle ahead, time to collision, lane occupations, etc.), and GPS data. To efficiently handle and process this massive dataset, we developed a data management framework for ingesting, processing, and managing the data. We received weekly data streams from GM, processed the trips, generated summary-level quality assurance/quality control (QA/QC) reports, identified outliers, and cleaned the data.

We augmented the data by performing map matching using HERE Maps API to extract road information (e.g., speed limits, traffic patterns, traffic signs, grade, etc.). We leveraged a VIN decoder to obtain vehicle model and trim-level information, and used an internal vehicle information database[20] to extract detailed vehicle specifications (e.g., vehicle mass, maximum engine power, frontal area, wheel details, etc.), and we integrated driver logs to identify the driver during each trip and the times at which drivers switched vehicles.

After ingestion, the processed and cleaned data allowed us to perform thorough analysis at different granularity levels.

## Data overview and distribution

Figure 5 shows the distribution of some selected variables that provide high-level information about the data. The distribution of trip distances is highly skewed—most trips are short ( < 200 km), with an average distance of around 25 km, and few trips are long-range. The

mean trip speed is approximately 50 km/h, with an average travel time of 22 minutes. The overall fleet-level fuel consumption is around 15 L/100 km (15.7 mpg). Fuel consumption values can vary depending on factors such as trip distance, time of year (e.g., short trips during cold seasons can result in extremely high fuel consumption), and vehicle type and model. The fuel consumption is determined using the "fuel injected rolling count" signal which is calculated in the vehicle's Engine Control Module (ECM) and broadcast over the vehicle's internal CAN network. The CAN network is monitored by an on-board data recorder that logs all signals continuously while driving.

Supplementary Table 2 provides a summary of the various vehicles included in the data. Each unique make/model/series may have multiple trim variants with different engine technologies and sometimes different fuel types (primarily gasoline and diesel). We used EPA Tier 3 87AKI certification fuel values for any fuel lower heating value and fuel density conversions to ensure consistent fuel comparisons. As the table shows, the fleet contains a few electric vehicles. The dataset also shows a relatively high use of automated driving technologies, with more than 35% of trips involving ACC usage. We do not believe that drivers were specifically instructed to use the automated features in their vehicles; instead, it is likely a result of their natural inclination to explore and try new technologies. Drivers typically used one vehicle type/model for extended periods, although multiple drivers may have used similar vehicles. A driver-vehicle matrix shows that drivers occasionally switch to different vehicle models throughout the year. The driver log enables us to track such events.

Map-matching GPS coordinates to road attribute data on HERE maps revealed that most trips are high functional class driving, typically consisting of local, short journeys with occasional highway usage. Trips generally include fewer than 10 traffic lights and a few stop signs.

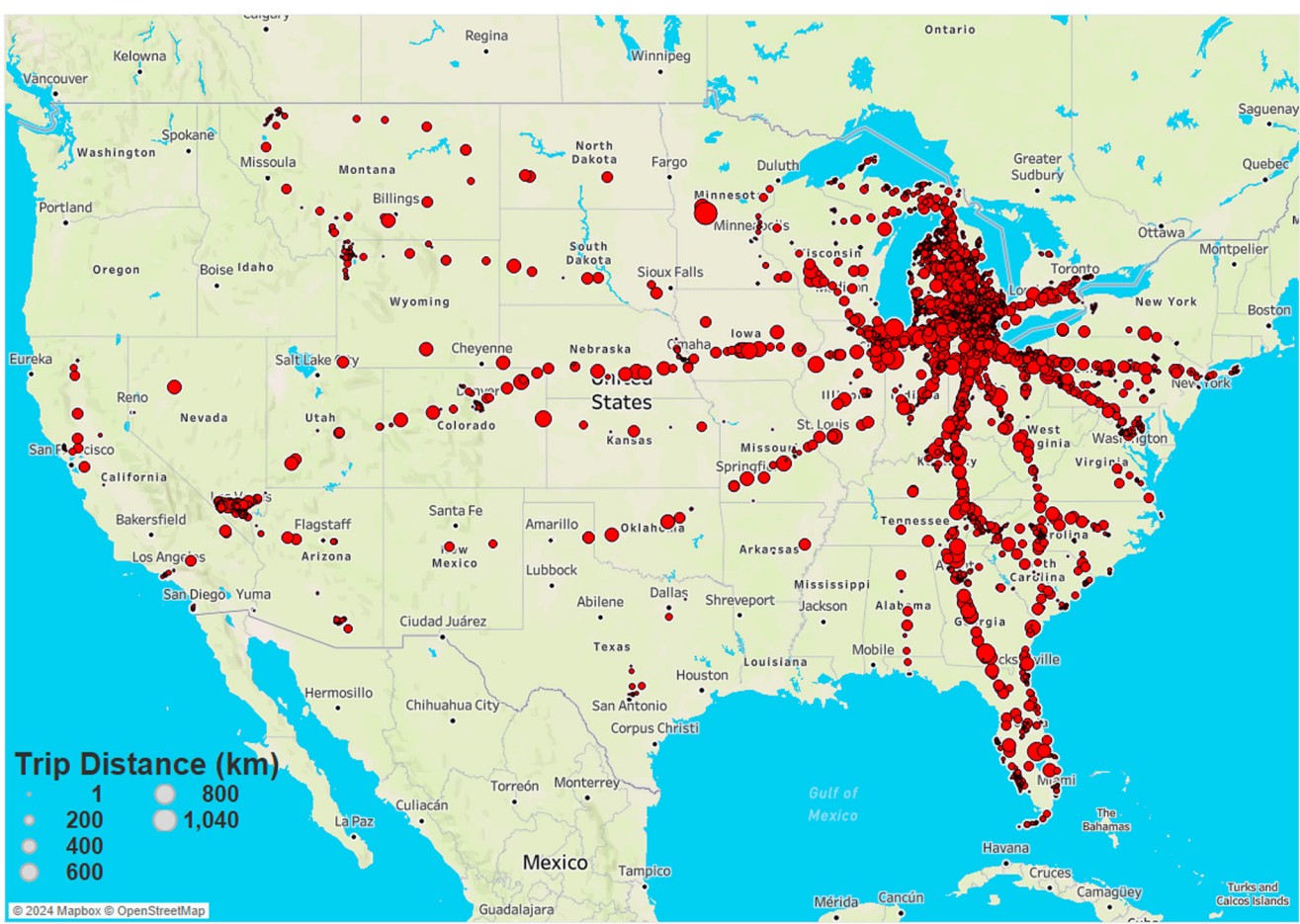

**Fig. 6 | Spatial distribution of recorded trips. Most trips are concentrated in the south-east Michigan area, with some cross-country trips.** Map data is available under the Open Database License(© OpenStreetMap).

Elevation changes during these trips are minimal, with delta elevation ranging between −200 m and 200 m. Vehicles used in this study have been equipped with GPS antennas that report elevation as well as lat/long coordinates. Given the potential for noisy GPS signals, these elevation data were cross-referenced with our map-matched elevation data from HERE maps to identify and eliminate potential anomalies.These are used to calculate elevation delta between different points in a route. That is, in a given trip or segment, the elevation delta is simply the difference in elevation between the two end points. Figure 6 provides a map view of the areas in the United States where the majority of trips occurred. As shown, most trips are concentrated in the southeast Michigan region, with some cross-country trips also taking place. As noted before, trips span a period of over a year of data collection; as a result, weather and ambient temperature levels span the range of conditions experienced in the lower 48 states in a typical calendar year. It was a priority to capture a wide range of temperatures and weather conditions in the data set, as weather can affect ACC engagement probability in two main ways. First, drivers may engage ACC at different rates depending on weather conditions - some may be more comfortable controlling the vehicle themselves in rainy/snowy conditions, while others with lower confidence or less driving experience may find the assistance of ACC to be helpful in inclement weather. Secondly, GM's ACC system cannot be engaged when the forward camera is obstructed, so system operation may be inhibited during severe precipitation events.

Finally, an in-depth analysis of driver and trip aggressiveness (which resulted in its own manuscript[21]) considered factors such as acceleration energy, jerk energy, and trip-level standard deviation metrics to gain a better understanding of trip and driver profiles in relation to the percentage of ACC utilization during trips. These exploratory and descriptive analyses provided valuable insights and informed the subsequent statistical design to accurately model the fleet.

## Vehicle characteristics

It is important to acknowledge that all vehicles in the study were produced by General Motors. While this is a limitation of the study, we do not expect that the inclusion of vehicles from other manufacturers would substantively change the results of the study, for a few reasons. First, although each automotive manufacturer has its own control algorithms for longitudinal and lateral control in cruise, we do not expect major differences in behavior because the high-level goals of these cruise systems are identical. The vehicles are programmed to maintain a single set speed unless traffic ahead requires them to slow down. While a vehicle is present ahead, driven vehicles are programmed to maintain a specified gap distance. Classical controls techniques (e.g., PI control) are employed nearly universally for maintaining open-road cruise speed and gap distance to the vehicle ahead. With regards to lateral control, some regions have unique regulations related to automatic lateral control that may on the surface imply differentiated cruise control performance between regions. However, in practice, the strategy for complying with UN Regulation 79 is likely identical across major automakers - vehicles remain under a prescribed lateral acceleration threshold by detecting upcoming curves in the roadway and slowing down in advance where necessary. Lastly, General Motors produces a wide variety of vehicles with

different engine sizes, transmission types, masses and body styles. The full breadth of the GM portfolio (including performance cars, sedans, crossovers, pickup trucks and full-size SUVs) was leveraged in this study, which approximates the composition of vehicles on North American roadways very well.

## Modeling techniques

The linear mixed-effect model was selected as the preferred technique for our analysis, as it is particularly suited to situations where the data have a nested structure, with observations grouped by certain factors. We accounted for the hierarchical structure of the data by considering random effects for vehicle type (VehicleType) and driver identification (DriverID), which are modeled as a representation of the variability associated with the grouping factors. Our model included eight fixed effects to be the variables of interest: inverse trip average vehicle speed in min/km (veh_spd_meanI), adaptive cruise control engagement (ACC_engaged_cat) as a binary variable, trip maximum vehicle speed in km/h (veh_spd_max), inverse trip distance covered in 1/km (dist_covered_kmI), trip-level vehicle acceleration energy in m²/s³ (veh_accel_nrg), trip elevation change in meters (elev_delta), trip average ambient temperature in degrees Celsius (amb_temp), and trip average engine temperature in degrees Celsius (eng_temp). These variables were included as covariates to control for their potential influence on fuel consumption while covariate transformations are meant to preserve a linear structure and ensure normality of model residuals.

The choice of a linear mixed effect model for this study allowed us to isolate the treatment effect of ACC engagement on fuel consumption as well as other covariates while controlling for the potential influence of VehicleType and DriverID in a cross-factored way. Note that, per our exploratory analysis, all variables included in this model exhibit fairly linear relationships with fuel – provided certain covariate transformation and under multiple controls. Normality of residuals and other model diagnostics revealed good model fit and no model assumption violation. Equation (3) represents the linear mixed effect model for the study:

$$
\begin{aligned}
\text{FuelCons}_i &\sim \mathcal{N}\left(\mu, \sigma^2\right) \\
\mu &= \alpha_{j[i],\,k[i]} + \beta_{1,\,k[i]} \times \text{veh\_spd\_meanI} \\
&\quad + \beta_{2,\,k[i]} \times \text{ACC\_engaged\_cat\_TRUE} \\
&\quad + \beta_3 \times \text{veh\_spd\_max} + \beta_4 \times \text{dist\_covered\_kmI} \\
&\quad + \beta_5 \times \text{veh\_accel\_nrg} + \beta_6 \times \text{elev\_delta} \\
&\quad + \beta_7 \times \text{amb\_temp} + \beta_8 \times \text{eng\_temp} \\
\alpha_j &\sim \mathcal{N}\left(\mu_{\alpha_j}, \sigma^2_{\alpha_j}\right), \quad \text{for DriverID } j = 1, \ldots, J \\
\begin{pmatrix} \alpha_k \\ \beta_{1k} \\ \beta_{2k} \end{pmatrix} &\sim \mathcal{N}\left( \begin{pmatrix} \mu_{\alpha_k} \\ \mu_{\beta_{1k}} \\ \mu_{\beta_{2k}} \end{pmatrix}, \begin{pmatrix} \sigma^2_{\alpha_k} & \rho_{\alpha_k,\beta_{1k}} & \rho_{\alpha_k,\beta_{2k}} \\ \rho_{\beta_{1k},\alpha_k} & \sigma^2_{\beta_{1k}} & \rho_{\beta_{1k},\beta_{2k}} \\ \rho_{\beta_{2k},\alpha_k} & \rho_{\beta_{2k},\beta_{1k}} & \sigma^2_{\beta_{2k}} \end{pmatrix} \right) \\
&\quad \text{for VehicleType } k = 1, \ldots, K
\end{aligned}
\tag{3}
$$

The first line, $\text{FuelCons}_i \sim N(\mu, \sigma^2)$, indicates that the fuel consumption (FuelCons$_i$) for each observation $i$ is modeled as a normally distributed random variable with mean $\mu$ and variance $\sigma^2$.

The next few lines describe the fixed-effects part of the model:

a. $\alpha_{j[i],k[i]}$ represents the intercept term for each observation $i$, accounting for both the random effects of DriverID $j$ and VehicleType $k$.

b. $\beta_{1k[i]}$(veh_spd_meanI) captures the effect of the inverse average vehicle speed with the coefficient $\beta_{1k[i]}$ specific to the VehicleType $k$.

c. $\beta_{2k[i]}$(ACC_engaged_catTRUE) represents the treatment effect of engaging ACC, with the coefficient $\beta_{2k[i]}$ specific to the VehicleType $k$.

d. $\beta_3$(veh_spd_max), $\beta_4$(dist_covered_kmI), $\beta_5$(veh_accel_nrg), $\beta_6$(elev_delta), $\beta_7$(amb_temp), and $\beta_8$(eng_temp) represent the effects

of maximum vehicle speed, distance covered, vehicle acceleration energy, elevation change, ambient temperature, and engine temperature, respectively, with their corresponding fixed coefficients.

The line $\alpha_j \sim N(\mu_{\alpha_j}, \sigma^2_{\alpha_j})$ describes the random effects for DriverID $j$. The intercept term $\alpha_j$ follows a normal distribution with mean $\mu_{\alpha_j}$ and variance $\sigma^2_{\alpha_j}$.

The remaining lines describe the random effects for VehicleType $k$. The model includes random effects for the intercept term $\alpha_k$ and the two fixed-effect coefficients $\beta_{1k}$ and $\beta_{2k}$. These random effects follow a multivariate normal distribution with means $\mu_{\alpha_k}$, $\mu_{\beta_{1k}}$, and $\mu_{\beta_{2k}}$, and a covariance matrix that captures the variances ($\sigma^2_{\alpha_k}$, $\sigma^2_{\beta_{1k}}$, and $\sigma^2_{\beta_{2k}}$) and correlations ($\rho_{\alpha_k\beta_{1k}}$, $\rho_{\alpha_k\beta_{2k}}$, and $\rho_{\beta_{1k}\beta_{2k}}$) among these random effects.

## Macroscopic trip-level analysis

This section presents a macroscopic-level analysis that investigates the fuel consumption (FC) outcomes at the trip level while focusing on the impact of ACC engagement. We approached this as a study of counterfactuals, considering the potential outcomes of FC with ACC engaged and with ACC disengaged. To estimate the causal effect of ACC engagement on FC, we use the concept of average treatment effect (ATE).

The ATE is the average difference in outcome between the treated group (ACC engaged) and the control group (ACC disengaged) in a hypothetical situation in which we can control for all potential confounding factors. In our study, the treatment is the ACC engagement, and the ATE represents the average effect of ACC engagement on fuel consumption.

The fundamental challenge in estimating the ATE is that we observe only one state of ACC engagement at a given time in each trip, i.e., either ACC is engaged or it is disengaged, not both at once. In a hypothetical scenario, where each trip could be observed under identical conditions with both ACC states, we could directly compute the ATE. However, since this is not the case, we must rely on large samples to compute the ATE by comparing sample means, assuming that the sample mean differences generate an unbiased estimate of the ATE:

$$
\begin{aligned}
\text{ATE} &= \frac{1}{N}\sum_i \tau_i \\
&= \frac{1}{N}\sum_i \left[ Y_i(1) - Y_i(0) \right]
\end{aligned}
\tag{4}
$$

This is equivalent in Expectation to the following:

$$
\text{ATE} = \frac{1}{N}\sum_i Y_i(1) - \frac{1}{N}\sum_i Y_i(0)
\tag{5}
$$

Where $\tau_i$ is an individual trip $i$ treatment effect, and $Y_i(1)$, $Y_i(0)$ are respectively the potential outcomes of trip $i$ when ACC is engaged and counterfactually not engaged and where ACC engagement is defined as ACC being turned on at least once, regardless of the duration or frequency of its use during the trip. This equivalence is valid only if every trip has an equal chance of engaging ACC.

In our dataset, trips do not have equal probabilities of engaging ACC, leading to group-level biases. Ideally, random ACC assignment would result in a true ATE estimate, as the differences between trips would balance out, eliminating these biases. In our case, vehicles with ACC engaged typically exhibit better FC outcomes due to different driving profiles (e.g., longer trips, higher average speeds). The bias originates from the fact that, because of the kinds of trips in which ACC is engaged, the mean FC for vehicles that engaged ACC, had they not engaged it, would differ from the mean FC for vehicles that did not engage ACC. This omitted variable bias can be addressed by introducing control variables that account for differences between trips,

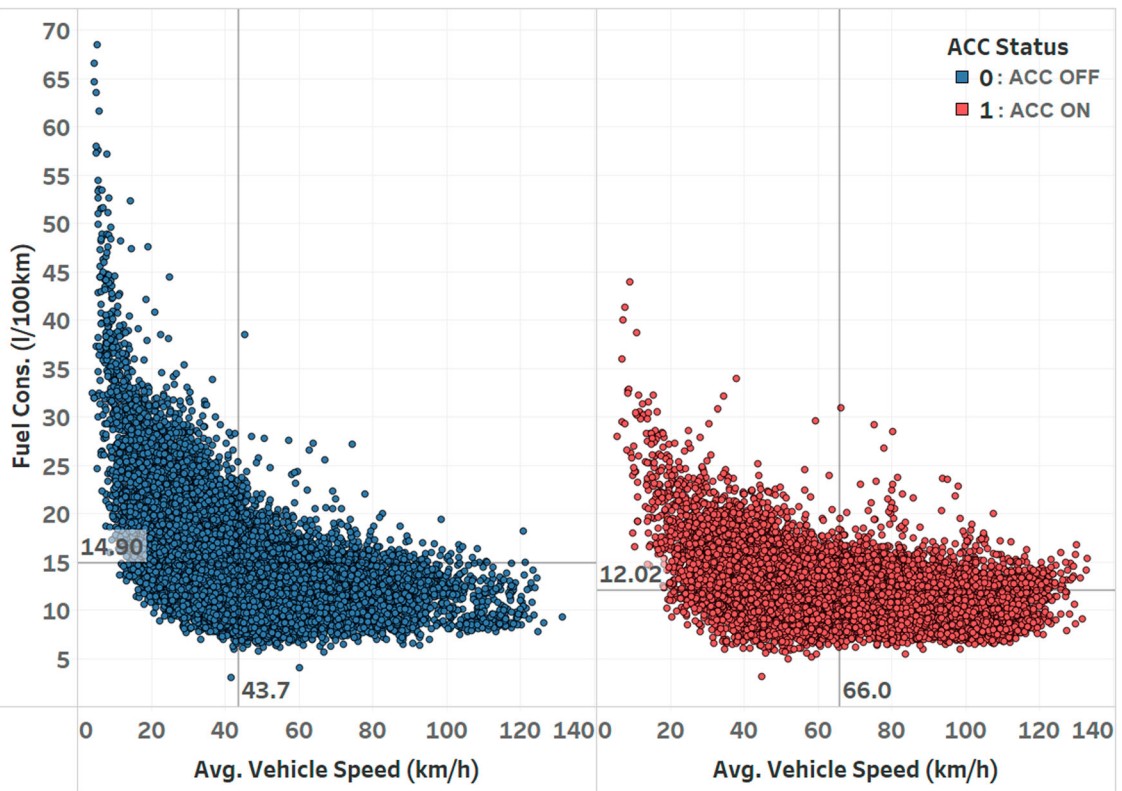

**Fig. 7 |** Trip-average fuel consumption vs. speed, separated by ACC (adaptive cruise control) engagement modes.

allowing for an apples-to-apples comparison and equalizing the two groups.

Without random ACC engagement, treatment and control groups are not random subsets of all trips. Engaging ACC is systematically related to reduced FC outcomes for reasons other than ACC itself. Our estimate of the ATE, the expected difference between trips with ACC ON and trips with ACC OFF, is equal to the ATE among trips with ACC ON (if it can be observed) plus the mean difference between trips that engaged ACC if they hadn't, and the mean of the group that did not engage ACC. This difference is typically zero if randomization is present. This is best presented by the following equation:

$$\mathbb{E}[Y_i(1)|G_i=1] - \mathbb{E}[Y_i(0)|G_i=0]$$
$$= \mathbb{E}[Y_i(1) - Y_i(0)|G_i=1] \qquad (6)$$
$$+ \left[\mathbb{E}[Y_i(0)|G_i=1] - \mathbb{E}[Y_i(0)|G_i=0]\right)]$$

where $G_i=1$ represents the group of trips in the data that actually engaged ACC, while potential outcomes $Y_i$ can be hypothetical. The first line is the expected difference between the ACC ON group and ACC OFF group, the second line is the ATE for the ACC ON group, and the last line represents the mean among the group that did engage ACC if they hadn't engaged ACC, which is different from the mean of the group that actually did not engage ACC. This last line captures the selection bias and would be nullified if $G_i=0$ and $G_i=1$ are similar. It is imperative to equalize the two groups in order to eliminate this bias term.

Because we want to study variation in ACC that is independent of FC, and because ACC engagement is not random among all trips, we need to equalize the trips by identifying variables that explain FC variation and that are related to ACC engagement likelihood. This process will help us to control for confounding factors and accurately estimate the impact of ACC on fuel consumption in real-world driving conditions.

## Statistical techniques for counterfactual analysis

In the following discussion, we leverage causal inference methods to estimate the effect of ACC on energy consumption. This approach involves identifying natural experiments in the data, such as trips where ACC was used versus others where ACC was not used. By comparing the fuel consumption of ACC-engaged trips to non-ACC trips, we could estimate the causal effect of ACC on energy consumption.

In our study, we consider two primary methodologies to analyze the impact of ACC on energy consumption: a controlled variable statistical model via regression and a propensity score matching (PSM). While PSM offers a more focused comparison by matching similar trips, there is complexity and sparsity in accurately matching and balancing trips. Conversely, regression analysis, less affected by these limitations, provides more comprehensive and applicable results, and thus, our findings will predominantly feature insights derived from the latter. In a multivariate approach (see section 2) we control for potential confounding factors that influence fuel consumption, such as trip distance, vehicle speed, driving conditions, vehicle type, and driver behavior, etc. (see section 5). We estimate the causal effects of the treatment (ACC engagement) on the outcome variable (fuel consumption) while controlling for these factors in order to isolate the effect of ACC engagement.

Compared to the previously used methodology, this approach (typically employed in observational studies) not only leads to a more accurate estimate of the true ACC impact on energy consumption, but also offers flexibility in terms of the functional forms and interactions between the selected variables, allowing for the exploration of more complex relationships between the treatment, the control variables, and the outcome.

## Controlled factors

We carefully designed a set of variables to control in order to achieve near group equality and normalize trips with and without ACC

**Table 3 | Summary table of debiasing variable selection**

| Variable | Effect | Explanation |
|---|---|---|
| Vehicle Type | 1st order | Large dispersion of FC over vehicle types; Unequal ACC utilization among vehicles due to location and trip types; Similar vehicle type can have multiple drivers. |
| Driver ID | 1st order | Large dispersion of FC over Driver IDs; Unequal usage of ACC among drivers. |
| Trip Average Speed | 1st order | Trip speed strong predictor of FC; Trip speed affects ACC engagement likelihood. |
| Trip Level Acceleration Energy | 1st order | Proxy for trip-level aggressiveness; A linear effect of acceleration energy is observed given a fixed driver, a fixed vehicle, and a fixed trip speed; Equalizes trip to a similar aggressiveness level without compromising the effect of ACC on total trip aggressiveness: only controls for strong trip-level traffic conditions. |
| Trip distance; Trip maximum speed; Trip delta elevation; Trip average engine temperature; Trip average ambient temperature | 2nd order | Provides additional trip equalization; Contributes to FC variability and correlates to ACC usage rate. |

engagement for comparison. This was achieved by learning the partial effects that each of the controlled variables has on fuel consumption and subtracting them to isolate the ACC effect. That is, we were trying to account for the effect of variables that we think have an impact on trip-level fuel consumption, learn that effect from the data and from the many examples, and remove their individual effects to isolate ACC impact.

Given the nature of this observational data, this variable selection endeavor boiled down to asking one fundamental question: What are the variables that explain FC variation that also could cause ACC to be engaged at the same time? That is, to understand the effects of ACC on FC, we have to think about not only the variables that cause FC variation but all the variables that cause ACC to be engaged. For example, Fig. 7 shows that average trip-level speed is a strong predictor of FC, but also is a source of bias for ACC engagement as ACC is engaged at higher average trip speed (66 km/h vs 42 km/h). Higher average speed leads to lower FC, as noted before. In fact, the figure shows a $1/x$ relationship, that is, fuel consumption over speed is a non-linear function. To model this relationship correctly within our linear model framework, we employed a covariate transformation by including an inverse speed term in the model to linearize the relationship. This approach preserves the linearity regression assumption and allows us to account for the non-linear nature of the fuel consumption-speed relationship.

The remaining selected variables are the results of careful data analysis. Primary factors such as the vehicle type and the driver ID are included, as well as first-order effects such as a speed and trip-level acceleration energy, defined as follows:

$$\frac{1}{T}\sum_{i=1}^{T}\left(\frac{\mathrm{d}v_i}{\mathrm{d}t_i}\right)^2 \mathrm{d}t_i \tag{7}$$

Note that acceleration energy is calculated at the trip level, where $T$ is a given trip length, $v_i$ is the speed at timestamp $i$, $t_i$ is the time signature at timestamp $i$, and some unit conversions are applied to provide $m^2/s^3$ unit values. We present the rest of the variables in Table 3 along with their descriptions.

It is important to note that the electrical power consumption of the control modules and sensors required for Super Cruise remains constant whether the system is engaged or not. The control module is designed to stay active and continuously process real-time sensor inputs, ensuring readiness for immediate engagement. Therefore, the difference in electrical power consumption between the engaged and disengaged states of Super Cruise is zero. Consequently, the energy consumption associated with the Super Cruise system is implicitly included in the fuel consumption data analyzed in this study and not controlled for.

## Data availability
The data that support the findings of this study are not publicly available due to privacy and confidentiality concerns. The dataset includes personally identifiable information (PII) in the form of GPS data, which traces the driving patterns of GM employees, including their commutes to and from home. As such, sharing the data publicly compromises the privacy of the individuals involved. Researchers interested in the dataset for collaborative projects or further analysis may contact the corresponding author to discuss potential data sharing under specific agreements that ensure the protection of privacy and confidentiality. Any data sharing would require approval and agreement from General Motors (GM) and would be contingent upon compliance with applicable privacy regulations and institutional guidelines.

## Code availability
The data used in this study was collected by General Motors using their proprietary data collection systems. Details on the specific software versions and technologies used for data collection are managed and maintained by GM. Additionally, the custom code and algorithms used for data processing and analysis and visualization were developed with Python 3.10, R 4.4.1 and Tableau 2024.1.5 software. Interested parties may contact the corresponding author for further information or potential collaboration, subject to approval.

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

## Acknowledgements

The work described was sponsored by the U.S. Department of Energy (DOE) Vehicle Technologies Office (VTO) under the Systems and Modeling for Accelerated Research in Transportation (SMART) Mobility Laboratory Consortium, an initiative of the Energy Efficient Mobility Systems (EEMS) Program. The following DOE Office of Energy Efficiency and Renewable Energy (EERE) managers played important roles in establishing the project concept, advancing implementation, and providing ongoing guidance: Erin Boyd, Prasad Gupte, Alexis Zubrow, Jacob Ward, and David Anderson. Additionally, we would like to express our gratitude to the team at General Motors for their invaluable support and contributions, particularly in the areas of data collection and delivery. Their expertise and dedication were instrumental in the success of this project. The submitted manuscript was created by UChicago Argonne, LLC, Operator of Argonne National Laboratory ("Argonne"). Argonne, a U.S. Department of Energy Office of Science laboratory, is operated under Contract No. DE-AC02-06CH11357. The U.S. Government retains for itself, and others acting on its behalf, a paid-up nonexclusive, irrevocable worldwide license in said article to reproduce, prepare derivative works, distribute copies to the public, and perform publicly and display publicly, by or on behalf of the Government. The Department of Energy will provide public access to these results of federally sponsored research in accordance with the DOE Public Access Plan.

## Author contributions

D.K., A.R., and A.M. conceptualized the study. A.M. developed the methodology. A.M., J.H., Y.Z. contributed to the software. M.Z. validated the findings. A.M. performed the formal analysis. M.Z. and A.M. conducted the investigation, while A.M. curated the data. A.M., M.Z., and J.H. wrote the original draft, with all authors reviewing and editing the manuscript. A.M. handled the visualization. D.K. and A.R. provided supervision and project administration. D.K. and A.R. also secured the funding for the project.

## Competing interests

The authors declare the following competing interests. This study was completed using proprietary, confidential data provided by General Motors through a Cooperative Research and Development Agreement (CRADA). All authors were involved in this CRADA and had access to the data.
