## [Transparent Peer Review file · Nature Communications]

Effect of Adaptive Cruise Control on Fuel Consumption in Real-World Driving Conditions

Corresponding Author: Mr Ayman Moawad

Version 0:

Reviewer comments:

Reviewer #1

(Remarks to the Author)

The manuscript presents research on the energy impacts of using Adaptive Cruise Control systems, comparing to human driving. A large amount of data is collected, in collaboration with a large manufacturer. Such a large amount of data can help clarify questions about the ACC characteristics, that are still discussed in the current literature. Additionally, the work presents value for the methodology used, that was designed, considering a possible bias in the areas where the ACC can/is activated. This bias has been ignored in some existing research works. The work is timely and certainly interesting. Such ADAS systems are the first iteration of real-world deployment of automated driving systems, even if it is low levels of automation, and investigations on those systems can help validate existing assumptions, or lead to new assumptions and further research. Moreover, the results of the study (2% increased fuel consumption with current systems), as well as the hypotheses regarding the different impacts, can be useful for researchers, practitioners, and regulators.

Some main comments:

1 The hypothesis of “drivers benefit from more flexibility compared to automated systems while in a cruising mode” is very important interesting. Further investigations to support it would be beneficial for the paper. The mechanism of flexibility bringing benefit to fuel consumption should be further explained, and if possible, evidence of this effect on the observed data should be found. The same could be attempted for the hypothesis that ACC is more efficient when following another human driver.

2 SuperCruise uses a group of different sensors that observe the driver, the vehicle in front, vehicles in adjacent lanes, and may combine with map data. Is there any estimation of the additional energy consumption due to the sensors being active or the computational power needed to process the data and decide the control inputs? Is that part of energy consumption included on the results presented, or it could be additional to them?

3 Specifications on the ACC system or systems used can be helpful. The vehicles used are equipped with the SuperCruise technology, coupling ACC with automated lateral control. Are all the cases of ACC driving referring to SuperCruise, or there is also parts of a more traditional ACC system being activated? Is there a difference energy-wise between the SuperCruise and the more traditional ACC systems?

Further comments:

4 The authors mention that “While ACC has been shown to improve safety and reduce driver workload”. Those points have to be referenced. Moreover, regarding safety the reviewer believes there is still a discussion going on, and similarly to the energy issue more data and research may be necessary.

5 Is there any possibility that the data, as they are or modified for privacy or intellectual property reasons, could be made openly accessible? Information regarding e.g. reaction times, distance headways, ACC activation/deactivation, can be useful, not in the present line of research, but for other researchers. Making such data openly available can increase a lot the impact of the experimental campaign.

6 Does the weather affect the probability to enable ACC, or the energy consumption? Temperature is considered, but for example precipitation could be significant as well. In principle, the authors could be able to get data regarding the weather conditions.

7 The authors mentioned that humans “leverage coasting before an actual brake event, which may lead to potential downstream benefits in the same segment” The potential benefits are probably upstream and not downstream. There are some works in regarding human drivers “multi-anticipation” which is the ability to react to more than one vehicle downstream (e.g. <https://ieeexplore.ieee.org/document/1707455>), which seems very much related to what the authors are explaining here. The authors briefly mention this point, but claim that “In practice, we find that few human drivers put in this level of thought and effort to achieve an efficient ride”. Are there any data contradicting the assumption that human drivers do use multi-anticipation? Moreover, are the ACC systems under test able to collect information for more than one leading vehicle? Are they using this information in any case?

8 The authors mention that the data has been cleaned. Some more details on the specific processes used could be useful.

9 Figure 1 can be further clarified as it is not very easy to read all the relevant information.

10 An analysis of the trip purposes and also an in-depth analysis of driver and trip aggressiveness are mentioned, but they are not presented in the present manuscript. Where can those be found?

11 It is not clear how exactly the elevation delta is calculated.

12 What type of data are used to estimate the fuel consumption? Is it a type of OBFDM?

13 The authors mention that “On the other hand, experimental studies, although potentially able to produce more reliable conclusions, require more resources and variations and offer limited scope for generalizations.” The statement is valid, however, some more nuance can be added. In such experimental studies, the commonly has to be a trained driver, or at least, a driver aware of the experiment going on. Hence, getting unbiased, naturalistic data for human drivers is not well achieved, and is rather difficult to do. This human driver bias is partly assessed in the present paper. However, more details regarding the human drivers of the systems should be given, as this could be an important source of bias.

14 An important limitation of the study is that only GM vehicles are used. Some further discussion is necessary on the point. How can the performance of the GM systems be compared to similar systems coming from different manufacturers? Moreover, considering that the systems are tested in the US, there is a question regarding similar systems from the same manufacturer introduced in different regions, where there are different regulatory restrictions. E.g. Regulation 79, that mostly covers automatic lateral control, can have some effects on the longitudinal part of driving as well. It is possible that there is no difference between the ACC systems deployed in the US and other regions, but clarifying this is important for any reader.

15 The authors claim that “Production ACC systems have generally been designed with an emphasis on functionality and safety.” Regardless of the validity of the statement, it does not seem to be supported by the evidence presented in the current work. Moreover, it is not clear if this statement is necessary for the work.

Reviewer #2

(Remarks to the Author)

This work analyzes a large pool of vehicle trajectory observations to assess the impact of ACC systems on fuel consumption. This is a challenging task since the underlying conditions when the ACC is on or off can differ significantly (traffic, vehicle powertrain, driver, elevation, geometry, etc.). This work seeks answers using statistical methods at macro- and micro-scopic levels.

I find the study very topical and challenging and the results very interesting. It complements the literature that is limited for such a large amount of data. More details on the data quality and analysis should be given as this part is not transparent. The findings are interesting as well and the methodology is mostly clear. However, the presentation needs to be improved in my opinion to extract/present more neatly the main takeaways to the interested reader. Not all the findings have the same level of certainty/confidence due to the nature of the dataset/analysis. Part of the analysis is to filter out primary takeaways and this is somehow neglected. Secondary takeaways can be also discussed but the need for further analysis in future research should be also highlighted. At the same time, some findings are contradictive to the literature and this is something that needs better elaboration. Finally, even though the authors make a nice attempt to identify key findings, there are significant limitations to the study (traffic context, trip speeds, trip lengths, powertrains, and different ACC settings are only some of those). Therefore, I think that Section VI.F needs to be enhanced and highlighted.

More detailed comments and suggestions:

- Do the authors account for different ACC settings (spacings)? My motivation for this question is that based on the literature, shorter settings under unstable conditions would lead to significantly larger fuel consumption. Please elaborate.
- I would suggest using unified metrics (km, miles, meters, seconds, hours) across the manuscript.
- The distribution of trip speeds has a clear peak around 50km/h. It is known that the fuel consumption over speed is a non-linear function. Would this speed-bias in the data be problematic for the presented findings? Furthermore, how the above peak influence the results presented in Fig.3?
- In continuation of the previous comment the authors state “Note that, per our exploratory analysis, all variables included in this model exhibit fairly linear relationships with fuel”. Maybe I misunderstood but I don't think that mean speed and vehicle

acceleration are linearly correlated with fuel consumption, i.e. “veh spd meanl” and “veh accel nrg”. I would appreciate further elaboration on this.

- What is the contribution of Table I, i.e. what is the main takeaway for the reader? E.g. what is the engine model, i.e. LT4, LZ0? Could this information be provided in an appendix?
- Where can someone find the following analysis? “Further analysis (not presented here) revealed that most trips are high functional class driving, typically consisting of local, short journeys with occasional highway usage.”
- I wonder how much data is enough for the ATE analysis presented in section IV. Will the results remain consistent if we progressively remove data from 100% to e.g. 70%? A short discussion on this would be helpful in my opinion.
- Variables using underscores in their names as they are introduced make the context more difficult to follow. I would suggest creating a table with observed variables and unique abbreviations for each one. Then figures like Fig.1 could use those abbreviations. For example in Fig.1 the y-axis label is “% of Total Count of _trip_fc” and in Fig.3 it is “Fuel Cons. (l/100km) – agg”. Although clear in meaning the a lack consistency in notation. Other examples include “veh spd meanl”, “ACC engaged catTRUE”, “amb_temp”, “eng_temp”, “veh_accel_nrg”, etc.
- I am not sure about the main message of Section IV.E: Do ACC systems consume less on lower speeds, i.e. below 50km/h?
- What do the numbers in Fig. 6 mean? Did you, for example, detect only two instance of acceleration and 2 for braking? I guess not. Are these exemplary subfigures?
- Tables III, IV, V give too much information and it would be nice if the authors detected and presented only the most significant findings. Full tables can go to an appendix in my opinion. What is the number in parentheses? I feel they should be significantly simplified.
- Is the dataset publicly available?

Version 1:

Reviewer comments:

Reviewer #1

(Remarks to the Author)

The reviewer’s comments have been mostly addressed. The manuscript is almost ready for publication. Two minor comments:

A) On the rebuttal, the authors mention on point 7 that “The tested ACC systems track both the first and second vehicle ahead as separate objects. The position, velocity, and acceleration of these vehicles are used as separate, distinct inputs in the ACC system’s decision-making process. The system therefore has potentially the capability to react to events like a sudden slowdown occurring several vehicles ahead, even if the vehicle directly in front has not yet reacted. However, while ACC systems have this capability, their current effectiveness in utilizing this data to employ strategies that optimize fuel efficiency is not clear”.

The reviewer believes that this may be useful information, that could be added in the discussion.

B) In the previous notes, I failed to explain what is the Regulation 79 I referred to. It is not EU Regulation 79, but UN Regulation No. 79 (<https://unece.org/transport/documents/2023/10/working-documents/un-regulation-no-79-revision-5>). I apologize for that.

Reviewer #2

(Remarks to the Author)

I would like to thank the authors for carefully replying to my comments and remarks. I have only a couple of final comments and I think that the paper can be accepted for publication.

Authors’ reply to Comment 7: One minor comment is that statements like the following can be misleading as they present facts (or insights) without discussing the way these insights were investigated. I would therefore suggest removing this discussion or put it in the conclusion as preliminary findings for future research:

“Further analysis (not presented here) revealed that most trips are high functional class driving, typically consisting of local, short journeys with occasional highway usage.”

Authors’ reply to Comment 8: I would argue that this comment is not fully tackled. Although I agree with the authors’ argumentation below, I still believe that it is important to state whether the main findings are still observed when the dataset is randomly reduced by e.g. 20%. Of course, I understand that the error will increase with a smaller dataset, so no absolute numbers are necessary. Nevertheless, the main key findings about the energy behavior of ACC systems should be there for a slightly reduced number of observations.

“However, we can provide some general insights based on statistical principles. More data generally leads to better results in statistical analysis. As the sample size (N) increases, the standard error decreases, leading to more precise estimates and reduced uncertainty.”

Rebuttal document

Introduction

Thank you for the detailed feedback provided on our manuscript. We appreciate the time and effort the reviewers have invested in evaluating our work and offering constructive suggestions. We have carefully considered all comments and have made corresponding revisions to the manuscript, which we believe have significantly improved the quality and clarity of our research.

Revisions

In response to the reviewers' comments, we have made several significant changes to improve the clarity, depth, and overall quality of the manuscript. We have expanded the discussion on how driver flexibility can lead to improved fuel consumption, including explanations based on related studies and preliminary results from segment-level data analysis. This addresses the hypothesis that human drivers can achieve better fuel efficiency through more dynamic speed adjustments compared to automated systems.

To clarify the energy consumption of the SuperCruise system's sensors, we have explained that the system's energy usage remains constant regardless of whether it is engaged. This factor is implicitly included in our fuel consumption analysis, addressing concerns about additional energy consumption due to sensor activity. We have also provided detailed information about the ACC systems used in our study, distinguishing between traditional ACC and SuperCruise systems to ensure readers understand that the longitudinal control is identical in both cases.

In an effort to maintain focus on the study's core topic of energy consumption, we have removed references to ACC's impact on safety and driver workload. Additionally, we have added a discussion on how weather conditions, such as precipitation, could influence ACC engagement probability and energy consumption, acknowledging the potential impact of external environmental factors on our findings.

We have elaborated on the concept of multi-anticipation and its potential benefits for fuel efficiency, supported by references to relevant literature. This addresses the reviewers' comments on human drivers' ability to anticipate and react to multiple vehicles ahead. Furthermore, while maintaining brevity, we provided additional details on our data cleaning and preparation processes to ensure transparency in our methodology.

To improve the clarity of our figures and tables, we have enhanced the captions, standardized measurement units throughout the manuscript, and clarified variable names in tables. We have also expanded the discussion on limitations and future research

directions, emphasizing the need for further analysis on various factors such as traffic context, trip speeds, and powertrains.

We mentioned ongoing studies that aim to further investigate ACC's impact on energy consumption using different driver profiles and machine learning methods. To improve readability, we moved detailed tables containing GM-specific information to the appendix. Additionally, we provided a detailed explanation of how elevation delta is calculated using GPS and map-matched data.

Lastly, we added a discussion on the importance of data sufficiency for the Average Treatment Effect (ATE) analysis and the challenges in conducting such a study.

Below are Point-by-Point Responses.

Rebuttal to Reviewer #1 Comments

1. Flexibility and Fuel Consumption

Comment: The hypothesis of “drivers benefit from more flexibility compared to automated systems while in a cruising mode” is very important interesting. Further investigations to support it would be beneficial for the paper. The mechanism of flexibility bringing benefit to fuel consumption should be further explained, and if possible, evidence of this effect on the observed data should be found. The same could be attempted for the hypothesis that ACC is more efficient when following another human driver.

Response: Thank you for your valuable feedback and for highlighting the importance of the hypothesis regarding driver flexibility and its impact on fuel consumption. We appreciate your suggestion to further investigate and support this hypothesis.

In the revised manuscript, we have incorporated a more detailed discussion on the mechanisms through which flexibility in human driving can lead to improved fuel consumption. This section now includes an explanation based on the findings of a related study [19], which demonstrated that allowing speed variation within defined limits can improve fuel consumption by reducing the need for engine braking and limiting powertrain downshifts.

In this additional discussion, we attempt to support and explain the underlying mechanism. However, we acknowledge that a higher resolution and more microscopic level analysis of the data is needed to understand the dynamics and powertrain-level effects on fuel consumption at a lower level. While such microscopic-level analysis is beyond the scope of this paper, two upcoming studies are expanding the results of this paper by (1) investigating the energy-saving benefits of ACC against different driver profiles. By categorizing drivers from efficient to aggressive based on driving behavior metrics, we aim to understand whether certain driver behaviors can amplify or diminish ACC's energy-saving potential and (2) leveraging machine learning methods to model the relationship of vehicle dynamics to energy consumption with and without ACC at a microscopic level (sec-by-sec level), providing a deeper insight into the powertrain and fuel consumption dynamics under varying conditions. These two studies will shed light on the nuanced interactions between driving behaviors, ACC, and energy consumption.

Finally, we have analyzed segments of the data where human drivers and ACC systems are both cruising under similar conditions. Preliminary results indicate that human drivers tend to have lower fuel consumption in these segments. This segment level, while not exhaustive, provides a basis for the hypothesis that we plan to include and detail in future papers. Below is an example of a set of cruising segments binned over various speeds ranges (control effect of speed) for a fixed vehicle (control effect of vehicle) and driver (control effect of driver), where fuel consumptions is compared is the two ACC ON/OFF mode. We observe that fuel consumption is generally lower when ACC is OFF in the high speed ranges. In addition, we notice that the standard deviation of speed is higher when ACC is OFF in these speed bins, which means that human drivers tend to vary their speed more compared to the more constant speed maintained by ACC systems. This indicates that human drivers adjust their speeds more dynamically. This variability can lead to more efficient driving patterns, such as coasting or mild acceleration and deceleration, which can optimize fuel usage (accounting for grade, anticipation of traffic conditions, and other environmental factors, etc.)

Link to change in the paper: Please see the blue text in Section VI, Part A (Open-Road Cruising).

2. Energy Consumption of Sensor Activity

Comment: SuperCruise uses a group of different sensors that observe the driver, the vehicle in front, vehicles in adjacent lanes, and may combine with map data. Is there any estimation of the additional energy consumption due to the sensors being active or the computational power needed

to process the data and decide the control inputs? Is that part of energy consumption included on the results presented, or it could be additional to them?

Response: Yes, GM electrical team studies have shown that the electrical power consumption remains constant regardless of whether SuperCruise is engaged or not. The control module for SuperCruise is designed to remain active and continuously process real-time sensor inputs, ensuring that the system is always ready for immediate engagement. Consequently, the difference in electrical power consumption between the engaged and disengaged states of SuperCruise is zero. Given that the power consumption of the control modules and sensors is constant and does not vary with the engagement of SuperCruise, this factor has been implicitly included in the results presented in our study. The fuel consumption data we analyzed reflects the overall energy usage of the vehicle during driving, which includes the baseline power consumption of the SuperCruise system.

This is an important clarification to make, hence we have included this detail in the revised paper.

Link to change in the paper: Please see the blue text in Section IV part C (Controlled factors)

3. ACC System Specifications

Comment: Specifications on the ACC system or systems used can be helpful. The vehicles used are equipped with the SuperCruise technology, coupling ACC with automated lateral control. Are all the cases of ACC driving referring to SuperCruise, or there is also parts of a more traditional ACC system being activated? Is there a difference energy-wise between the SuperCruise and the more traditional ACC systems?

Response: SuperCruise technology incorporates ACC for longitudinal control and adds automated lateral control to maintain the vehicle's position in the center of a lane and perform automated lane changes. The longitudinal control system in SuperCruise is identical to that of traditional ACC. Therefore, all instances of ACC driving in our study refer to the same type of longitudinal control, whether SuperCruise or traditional ACC is engaged. Some of the vehicles in our study were equipped with ACC only, while others had the full SuperCruise system. However, the longitudinal control for both systems is identical. This means that the comparison of fuel consumption between ACC ON and ACC OFF states applies uniformly across vehicles with either system. While there is a difference in electrical energy consumption between a vehicle equipped with standard ACC and one equipped with SuperCruise due to the additional modules, sensors, and components required for lateral control in SuperCruise, this difference is not a concern for our study. The power consumption remains constant in both ACC modes regardless of whether ACC or SuperCruise is engaged or disengaged. The energy consumption differences between the systems do not impact our comparative analysis of fuel consumption (as we are controlling for the vehicle and therefore the underlying ACC configuration and technology).

Link to change in the paper: We added a brief clarification in Section II, part B as a footnote.

4. Safety and Workload References

Comment: The authors mention that “While ACC has been shown to improve safety and reduce driver workload”. Those points have to be referenced. Moreover, regarding safety the reviewer

believes there is still a discussion going on, and similarly to the energy issue more data and research may be necessary.

Response: Thank you for this comment. Given that the focus of this manuscript is on ACC's impact on energy consumption rather than its implications for safety and driver workload, we have decided to remove these references to topics beyond the scope of this work.

Link to change in the paper: Removed reference. We hope this modification addresses your concern and keeps the manuscript focused on its core topic.

5. Data Accessibility

Comment: Is there any possibility that the data, as they are or modified for privacy or intellectual property reasons, could be made openly accessible? Information regarding e.g. reaction times, distance headways, ACC activation/deactivation, can be useful, not in the present line of research, but for other researchers. Making such data openly available can increase a lot the impact of the experimental campaign.

Response: Thank you for your suggestion regarding the possibility of making the data openly accessible. We appreciate the potential value that such data could offer to the broader research community. However, the data cannot be distributed because it contains PII (personally identifiable information) in the form of GPS data. Many of these drivers are GM employees driving the cars to and from their homes. The authors are open to further conversation on this topic if it would be helpful.

Link to change in the paper: We have added a data availability statement at the end of the paper.

6. Weather Conditions and ACC Use

Comment: Does the weather affect the probability to enable ACC, or the energy consumption? Temperature is considered, but for example precipitation could be significant as well. In principle, the authors could be able to get data regarding the weather conditions.

Response: This is an excellent question, and we feel the following discussion could be a valuable addition to the paper.

Precipitation (and/or high winds) could affect ACC engagement probability in two main ways.

1) Driver behavior

a. Drivers may have a different tolerance for automated driving technology in inclement weather compared to clear conditions. It is possible that some drivers are more comfortable controlling the vehicle themselves (through the steering wheel / pedals) in rainy/snow conditions where quick reactions could be required to mitigate hydroplaning/skids or avoid collisions with other surrounding vehicles. In these cases, we would expect ACC engagement probability to drop in the presence of precipitation and/or high winds.

b. Other drivers with less driving experience or lower confidence may find increased driver assistance (e.g. from ACC) to be helpful in inclement weather conditions. In these

cases, we would expect ACC engagement probability to rise in the presence of precipitation and/or high winds.

2) Sensor-related system inhibits

a. In cases where the forward-facing camera is sufficiently obscured (either by dirt, debris, or precipitation) the GM SuperCruise system cannot be engaged. When the obstruction is clear (e.g. when precipitation stops), the system can be re-engaged. If inhibits are present, we expect ACC engagement probability to drop in the presence of precipitation.

While temperature is already considered in our analysis, incorporating additional weather data such as precipitation and wind conditions could provide a more comprehensive understanding of the factors influencing ACC engagement and energy consumption. We will explore the feasibility of obtaining and integrating weather condition data into our analysis in future work.

Link to change in the paper: Blue text in Section III.B (Data Overview and Distribution)

7. Human Drivers and Coasting Benefits

Comment: The authors mentioned that humans “leverage coasting before an actual brake event, which may lead to potential downstream benefits in the same segment” The potential benefits are probably upstream and not downstream. There are some works in regarding human drivers “multi-anticipation” which is the ability to react to more than one vehicle downstream (e.g. <https://ieeexplore.ieee.org/document/1707455>), which seems very much related to what the authors are explaining here. The authors briefly mention this point, but claim that “In practice, we find that few human drivers put in this level of thought and effort to achieve an efficient ride”. Are there any data contradicting the assumption that human drivers do use multianticipation? Moreover, are the ACC systems under test able to collect information for more than one leading vehicle? Are they using this information in any case?

Response: The confusing upstream/downstream language in the paper has been removed, in favor of the following statement which more directly states our hypothesis: “In braking situations, the penalty that ACC offers is less clear (+0.334 L/100 km); however, we hypothesize that human drivers are better able to leverage coasting before an actual brake event, which may lead to effective efficiency benefits as the nominal fuel consumption of a deceleration event is spread over a greater distance traveled”.

Regarding the concept of multi-anticipation, we acknowledge the ability of human drivers to react to more than one vehicle downstream. The work referenced (<https://ieeexplore.ieee.org/document/1707455>) provides valuable insights into this capability. While our hypothesis is based on the observation that not all drivers consistently employ such strategies to optimize fuel efficiency, we recognize that data supporting the prevalence of multi-anticipation among human drivers would be beneficial. We will consider this in our future research.

The tested ACC systems track both the first and second vehicle ahead as separate objects. The position, velocity, and acceleration of these vehicles are used as separate, distinct inputs in the ACC system’s decision-making process. The system therefore has potentially the capability to

react to events like a sudden slowdown occurring several vehicles ahead, even if the vehicle directly in front has not yet reacted. However, while ACC systems have this capability, their current effectiveness in utilizing this data to employ strategies that optimize fuel efficiency is not clear. As object sensing and detection technologies improve over time, there will only be greater opportunity to anticipate changes in advance and recalculate optimized driving behaviors more effectively than humans.

Link to change in the paper: Blue text in Section V. part B (results) + added multi-anticipation reference.

8. Data Cleaning Details

Comment: The authors mention that the data has been cleaned. Some more details on the specific processes used could be useful.

Response: We agree that transparency in data preparation is important. While detailed data cleaning processes are essential to ensure the integrity of our analysis, we are constrained by the paper's length limitations. We feel that Section III part A (Data Collection and Management) section provides the necessary information and an overview of the process without going into details.

Link to change in the paper: Please refer to Section III, part A.

9. Clarity of Figure 1

Comment: Figure 1 can be further clarified as it is not very easy to read all the relevant information.

Response: We acknowledge that the figure contains a significant amount of information, which may affect its readability. However, given the constraints of paper length and the need to provide a comprehensive overview of key trip-level data distribution statistics, we believe that the current format is the most efficient way to convey this information. We made several attempts to update and improve the figure without success, and we feel it is an important "data summary" figure. Figure 1 includes four histograms that illustrate important trip-level data distribution statistics related to trip distance, speed, travel time, and fuel consumption. These histograms are crucial for providing an overview of the dataset, allowing readers to understand the range and distribution of the variables analyzed in the study.

To address your concern, we propose to enhance the caption to summarize the key insights from the visual. We hope this modification improves the clarity and usefulness of Figure 1.

Link to change in the paper: Enhanced caption detailing key insights.

10. Analysis of Trip Purposes

Comment: An analysis of the trip purposes and also an in-depth analysis of driver and trip aggressiveness are mentioned, but they are not presented in the present manuscript. Where can those be found?

Response: Our group initially planned to include the driver-specific analysis in this paper. However, the scope of this analysis grew significantly, leading us to decide that these findings warranted a separate, dedicated paper ("Eco or Ego? Dissecting Driver Styles in the Adaptive

Cruise Control Fuel Consumption Saga," in preparation). Although this study is not yet complete, we are pleased to share the preliminary findings with the reviewers. We have also included a mention of this forthcoming paper in the body of the manuscript to provide additional clarity for the readers.

Link to change in the paper: Added reference to forthcoming paper.

11. Calculation of Elevation Delta

Comment: It is not clear how exactly the elevation delta is calculated.

Response: The vehicles used in this study are equipped with GPS antennas that report elevation along with latitude and longitude coordinates. These GPS-reported elevation signals are used to calculate the elevation delta between different points in a route. Given the potential for noisy GPS signals, these elevation data were cross-referenced with our map-matched elevation data from HERE maps to identify and eliminate potential anomalies. HERE maps is a highly reliable data source, widely used in the industry for its accurate and comprehensive geographic data.

In a given trip or segment, the elevation delta is simply the difference in elevation between the two end points.

Link to change in the paper: Blue text in Section III.B (Data Overview and Distribution).

12. Fuel Consumption Data Type

Comment: What type of data are used to estimate the fuel consumption? Is it a type of OBFCEM?

Response: The fuel consumption is determined using the “fuel injected rolling count” signal which is calculated in the vehicle’s Engine Control Module (ECM) and broadcast over the vehicle’s internal CAN network. The CAN network is monitored by an on-board data recorder that logs all signals continuously while driving.

The “fuel injected rolling count” signal is a pre-integrated volume of fuel that has passed through the injectors, reported in liters. We chose to use a signal that is integrated on-board, because all data was transmitted at low temporal resolution (1 Hz). This allowed us to avoid the numerical integration error that could result from a Riemann or trapezoidal sum on an instantaneous fuel flow signal that can vary dramatically over the course of a 1-second reporting interval.

Link to change in the paper: We have clarified this in blue text in Section III.B (Data Overview and Distribution).

13. Human Driver Bias and Experimental Data

Comment: The authors mention that “On the other hand, experimental studies, although potentially able to produce more reliable conclusions, require more resources and variations and offer limited scope for generalizations.” The statement is valid, however, some more nuance can be added. In such experimental studies, the commonly has to be a trained driver, or at least, a driver aware of the experiment going on. Hence, getting unbiased, naturalistic data for human drivers is not well achieved, and is rather difficult to do. This human driver bias is partly assessed in

the present paper. However, more details regarding the human drivers of the systems should be given, as this could be an important source of bias.

Response: We appreciate the reviewer’s insight on this topic and have incorporated the additional nuance into the body of the paper.

Link to change in the paper: Blue text in Section I.C (Literature Review).

14. Limitations with GM Vehicles

Comment: An important limitation of the study is that only GM vehicles are used. Some further discussion is necessary on the point. How can the performance of the GM systems be compared to similar systems coming from different manufacturers? Moreover, considering that the systems are tested in the US, there is a question regarding similar systems from the same manufacturer introduced in different regions, where there are different regulatory restrictions. E.g. Regulation 79, that mostly covers automatic lateral control, can have some effects on the longitudinal part of driving as well. It is possible that there is no difference between the ACC systems deployed in the US and other regions, but clarifying this is important for any reader.

Response: We agree that further discussion on this topic is important and have added the following passage to the manuscript:

“It is important to acknowledge that all vehicles in the study were produced by General Motors. While this is a limitation of the study, we do not expect that the inclusion of vehicles from other manufacturers would substantively change the results of the study, for a few reasons. First, although each automotive manufacturer has its own control algorithms for longitudinal and lateral control in cruise, we do not expect major differences in behavior because the high-level goals of these cruise systems are identical. The vehicles are programmed to maintain a single set speed unless traffic ahead requires them to slow down. While a vehicle is present ahead, driven vehicles are programmed to maintain a specified gap distance. Classical controls techniques (e.g., PI control) are employed nearly universally for maintaining open-road cruise speed and gap distance to the vehicle ahead. With regards to lateral control, some regions (for example, the EU) have unique regulations related to automatic lateral control that may on the surface imply differentiated cruise control performance between regions. However, in practice, the strategy for complying with EU Regulation 79 is likely identical across major automakers –vehicles remain under a prescribed lateral acceleration threshold by detecting upcoming curves in the roadway and slowing down in advance where necessary. Lastly, General Motors produces a wide variety of vehicles with different engine sizes, transmission types, masses and body styles. The full breadth of the GM portfolio (including performance cars, sedans, crossovers, pickup trucks and full-size SUVs) was leveraged in this study, which approximates the composition of vehicles on North American roadways very well.”

We hope this modification provides the necessary context and addresses the concern about the study's limitations and the applicability of its findings across different manufacturers and regions.

Link to change in the paper: New Section III.C (Vehicle Characteristics).

15. ACC Design Emphasis

Comment: The authors claim that “Production ACC systems have generally been designed with an emphasis on functionality and safety.” Regardless of the validity of the statement, it does not seem to be supported by the evidence presented in the current work. Moreover, it is not clear if this statement is necessary for the work.

Response: Thank you for this comment. This statement has been removed from the paper to improve conciseness and readability.

Link to change in the paper: Removed

Rebuttal to Reviewer #2 Comments

1. General Assessment of the Study

Comment: The work analyzes a large pool of vehicle trajectory observations to assess the impact of ACC systems on fuel consumption. This is a challenging task since the underlying conditions when the ACC is on or off can differ significantly (traffic, vehicle powertrain, driver, elevation, geometry, etc.). This work seeks answers using statistical methods at macro- and micro-scopic levels. The study is very topical and challenging, and the results are interesting. It complements the literature that is limited for such a large amount of data. More details on the data quality and analysis should be given as this part is not transparent. The findings are interesting as well and the methodology is mostly clear. However, the presentation needs to be improved in my opinion to extract/present more neatly the main takeaways to the interested reader. Not all the findings have the same level of certainty/confidence due to the nature of the dataset/analysis. Part of the analysis is to filter out primary takeaways and this is somehow neglected. Secondary takeaways can be also discussed but the need for further analysis in future research should be also highlighted. At the same time, some findings are contradictory to the literature and this is something that needs better elaboration. Finally, even though the authors make a nice attempt to identify key findings, there are significant limitations to the study (traffic context, trip speeds, trip lengths, powertrains, and different ACC settings are only some of those). Therefore, I think that Section VI.F needs to be enhanced and highlighted.

Response: Thank you for your comprehensive and constructive feedback on our manuscript. We appreciate your acknowledgment of the challenging nature of our work and the value of our findings. First, we would like to provide a general response to the points you raise in your general assessment of the study before addressing them point by point:

While we recognize the importance of detailing data quality and analysis, the main focus of our manuscript is on presenting the results of our study. Due to space constraints, we have not detailed the data quality checks and processing steps extensively. However, we assure you that rigorous data cleaning and verification processes were employed, as briefly mentioned in our

methodology section. We believe that elaborating on these steps, while valuable, would detract from the primary focus on our findings.

We also agree that transparency in data preparation is important. While detailed data cleaning processes are essential to ensure the integrity of our analysis, we are constrained by the paper's length limitations. We feel that Section III part A (Data Collection and Management) section provides the necessary information and an overview of the process without going into detail.

Regarding the presentation and extraction of main takeaways, we appreciate your suggestion to improve the presentation of our key insights. We have made several structural changes and additions to the manuscript to enhance clarity and highlight the main findings. We hope these improvements will make the key insights more accessible to the interested reader.

Finally, we acknowledge the significant limitations you mentioned, such as traffic context, trip speeds, trip lengths, powertrains, and different ACC settings. These factors are mostly controlled for in our analysis at both the trip level and the segment level analysis (macroscopic level). We have listed the remaining uncertainties in the limitations section, discussing other potential confounders that could impact the results. We agree that Section VI.F should be enhanced, and we made revisions to better highlight these limitations and the need for further research.

We hope these modifications address your concerns and improve the overall clarity and impact of our manuscript.

Link to change in the paper: Please refer to various parts of the paper, specifically Section III for data, the result sections, and Section VI for the discussion and limitations.

2. ACC Settings and Fuel Consumption

Comment: Do the authors account for different ACC settings (spacings)? My motivation for this question is that based on the literature, shorter settings under unstable conditions would lead to significantly larger fuel consumption. Please elaborate.

Response: We acknowledge the importance of this factor, as literature indicates that shorter settings under unstable conditions can lead to significant fuel consumption variation.

ACC settings, including gap and other parameters, are indeed available in the data signals we collected. However, this parameter was not taken into account in this study. We chose not to include it to avoid complicating the study, the analysis, and the presentation of the findings. Our primary focus was to provide a generally clear and comprehensible analysis of ACC's impact on fuel consumption. We recognize that including ACC settings could have enriched the analysis by accounting for additional variability in the extracted effect. We have acknowledged this limitation in our manuscript.

Link to change in the paper: Please see the blue text in Section VI, Part G (Limitations of the Study).

3. Consistency of Measurement Units

Comment: I would suggest using unified metrics (km, miles, meters, seconds, hours) across the manuscript.

Response: Thank you for this comment. Units have been standardized across the manuscript to enhance clarity and readability. The one exception is the average fuel economy figure for the fleet in the data overview, which is given in two units, both L/100km (which is used in the remainder of the manuscript) and mpg, for those who may be unfamiliar with the conversion.

Link to change in the paper: Units change all throughout the paper.

4. Speed Distribution and Fuel Consumption Analysis

Comment: The distribution of trip speeds has a clear peak around 50km/h. It is known that the fuel consumption over speed is a non-linear function. Would this speed-bias in the data be problematic for the presented findings? Furthermore, how the above peak influence the results presented in Fig.3?

Response: The clear peak around 50 km/h in the distribution of trip speeds is reflective of typical urban and suburban driving conditions, which are common in our dataset. We agree that this speed bias could influence the overall fuel consumption results since fuel efficiency can vary significantly with speed. To address this potential bias, we performed our analyses by controlling for speed and other confounding factors. By removing the effect of speed, we aimed to mitigate the impact of any single speed on our overall findings. This approach allows us to isolate the effect of ACC on fuel consumption more accurately across different driving conditions.

As you mentioned, fuel consumption exhibits a non-linear relationship with speed. In Figure 3, this is shown by a roughly $1/x$ relationship. To model this relationship correctly within our linear model framework, we employed a covariate transformation by including an inverse speed term in the model to linearize the relationship. This approach allows us to account for the non-linear nature of the fuel consumption-speed relationship and mitigate any bias introduced by the peak in the speed distribution.

To clarify this point, we added a brief explanation in the controlled factors subsection about how we accounted for the non-linear relationship between speed and fuel consumption.

Link to change in the paper: Please refer to Section IV.C.

5. Clarification on Variable Relationships

Comment: In continuation of the previous comment the authors state “Note that, per our exploratory analysis, all variables included in this model exhibit fairly linear relationships with fuel”. Maybe I misunderstood but I don’t think that mean speed and vehicle acceleration are linearly correlated with fuel consumption, i.e. “veh spd mean1” and “veh accel nrg”. I would appreciate further elaboration on this.

Response: Thank you for your follow-up comment regarding the linear relationships between variables and fuel consumption. We believe there may have been a misunderstanding, and this is an important point to clarify on our end. The linear relationships we refer to are indeed those observed after applying appropriate transformations to the variables. Our exploratory analysis was instrumental in identifying the best transformations and engineering the most effective features for our model, ensuring that the relationships with fuel consumption are linear post-transformation,

and that the variables exhibit strong effect i.e., have good explanatory power with respect to fuel consumption variability.

For example, we recognize that mean speed is not inherently linearly correlated with fuel consumption. However, through our exploratory analysis, we identified that including the inverse speed term allows us to model these relationships linearly within our regression framework. The term “veh spd meanl” represents the inverse of vehicle speed.

Link to change in the paper: We have added an emphasis on how the linearity is conditioned on certain covariate transformation and under multiple controls in Section IV.D.

6. Relevance of Table I

Comment: What is the contribution of Table I, i.e. what is the main takeaway for the reader? E.g. what is the engine model, i.e. LT4, LZ0? Could this information be provided in an appendix?

Response: Thank you for this comment. We can relocate this table to the Appendix to improve conciseness and readability, particularly considering the quantity of GM-specific information included in the table (e.g. internal engine codes like LT4 and LZ0). The point was to show that similar make, model, series can still have different engine technologies (e.g. gasoline or diesel version of the same vehicle). This might not present sufficiently useful information for most readers to be included in the body of the text, but can still be available in the Appendix for readers that are interested in the specific makeup of the fleet.

Link to change in the paper: Table can and will be moved to appendix. We have added and linked descriptions of the engines in the footnote in the caption. Please advise.

7. Trip Type Analysis

Comment: Where can someone find the following analysis? “Further analysis (not presented here) revealed that most trips are high functional class driving, typically consisting of local, short journeys with occasional highway usage.”

Response: The analysis referred to was purely exploratory. These internal analyses were conducted to gain a better understanding of the dataset and to inform our approach to the main study. They were not planned to be shared publicly and were not included in the manuscript due to space constraints and the focus on presenting the core findings of our research. We are constrained by the limitations on paper length and scope, which necessitate a focus on the primary analyses and results directly relevant to our study’s objectives. Including all exploratory and supporting analyses would detract from the clarity and conciseness of the main findings we aim to present.

We hope that if these data descriptive analyses are of interest, we can consider publishing them in a separate manuscript in the future.

Link to change in the paper: NA

8. Data Sufficiency for ATE Analysis

Comment: I wonder how much data is enough for the ATE analysis presented in section IV. Will the results remain consistent if we progressively remove data from 100% to e.g. 70%? A short discussion on this would be helpful in my opinion.

Response: The sufficiency of data for the ATE analysis is an important and insightful question, especially for future researchers that would be interest in conducting such study in the future. We did not conduct a specific data sufficiency study to determine the minimum amount of data needed for consistent ATE results. However, we can provide some general insights based on statistical principles.

More data generally leads to better results in statistical analysis. As the sample size (N) increases, the standard error decreases, leading to more precise estimates and reduced uncertainty. This principle, known as statistical power, is crucial for detecting small effects, such as the impact of ACC on fuel consumption. When the effect signal in the data is small, a significant amount of data is needed to overcome the noise. Additionally, statistical consistency implies that as the sample size increases, the bias in the estimations is reduced, providing more reliable results.

The effect of ACC on fuel consumption is relatively small, necessitating a significant amount of data to detect the signal amid the noise. But also, we control for many variables in our analysis, effectively slicing the data space across multiple dimensions. To ensure statistical significance and meaningful results in this high-dimensional space, a large number of data points are needed. Without sufficient data, the hypercubes within this multidimensional space would lack enough points to draw reliable conclusions.

While we did not specifically test the consistency of results by progressively removing data from 100% to 70%, we acknowledge the importance of such an analysis. It is worth noting that conducting a data sufficiency study is not as simple as it may seem, as there are various ways the data could be sampled, including random sampling, cluster sampling, or stratified sampling. Each of these methods can introduce different biases and complexities, requiring a sophisticated study design to offer relevant recommendations. In our case, we are dealing with a natural experiment from a purely observational study, making it challenging to predict how the results might differ under alternative sampling methods.

In our data section, we have presented everything we know about this natural experiment, including diverse vehicles, drivers, different locations, and varying ACC engagement, without controlling for the outcomes of any of these factors. Future research could include a well-crafted data sufficiency study to determine the minimum data requirements for robust ATE analysis.

Link to change in the paper: We offer to discuss this interesting point in our discussion Section VI.H “limitation of the study”. We have significantly expanded and reorganized this section. Please refer to blue text.

9. Standardization of Variable Naming

Comment: Variables using underscores in their names as they are introduced make the context more difficult to follow. I would suggest creating a table with observed variables and unique abbreviations for each one. Then figures like Fig.1 could use those abbreviations. For example in Fig.1 the y-axis label is “% of Total Count of _trip_fc” and in Fig.3 it is “Fuel Cons. (l/100km) – agg”.

Although clear in meaning there is a lack of consistency in notation. Other examples include “veh_spd_mean1”, “ACC_engaged_catTRUE”, “amb_temp”, “eng_temp”, “veh_accel_nrg”, etc.

Response: Thank you for this comment. We have added descriptive titles to all variable names in all tables in the manuscript.

Link to change in the paper: Variable names were changed and clarified all throughout the paper.

10. Low-Speed ACC Fuel Consumption

Comment: I am not sure about the main message of Section IV.E: Do ACC systems consume less on lower speeds, i.e. below 50km/h?

Response: Section IV.E aims to analyze the interaction between ACC engagement and average trip speed, focusing on how the effect of ACC on fuel consumption varies with different trip speeds. Our findings indicate that the impact of ACC on fuel consumption is not uniform across all speed ranges. The key points are:

- Overall, ACC engagement results in a slight increase in fuel consumption (+0.26 L/100 km) when considering all trip speeds (from the section before Section IV.D)
- By introducing an interaction term between ACC engagement and average trip speed, we observed that the effect of ACC on fuel consumption depends on the speed of the trip.
- Specifically, our analysis showed that ACC systems tend to consume less fuel at lower speeds (below 50 km/h). This is indicated by the interaction term, which suggests a fuel consumption benefit for ACC at these lower speeds. The analysis identified a speed threshold around 50 km/h. Below this threshold, ACC engagement is associated with reduced fuel consumption, whereas above this threshold, ACC tends to increase fuel consumption.
- This finding aligns with the understanding that ACC is more efficient in certain driving conditions, particularly in urban and suburban environments where average speeds are typically lower. At higher speeds, particularly on highways, the rigid control of ACC to maintain a constant speed might lead to less efficient fuel usage compared to human drivers who may adjust their speed more flexibly.

To clarify this point and enhance the understanding of Section IV.E we have added a paragraph detailing the takeaway of this section. We hope this modification improves the section clarity.

Link to change in the paper: Please refer to blue text in Section IV.E.

11. Interpretation of Figure 6

Comment: What do the numbers in Fig. 6 mean? Did you, for example, detect only two instances of acceleration and 2 for braking? I guess not. Are these exemplary subfigures?

Response: The numbers in the subfigure titles refer to the number of detected events of a given type in a given trip. For example, BnA (26) means that 26 “Brake and Acceleration” events were detected in the displayed trip. The number of pure acceleration, pure braking, and creep events is quite limited throughout the dataset, because braking and acceleration maneuvers are typically located close to each other, creating “BnA” and “BSnA” type events. These are exemplary figures.

Link to change in the paper: Improved caption.

12. Simplification of Data Presentation

Comment: Tables III, IV, V give too much information and it would be nice if the authors detected and presented only the most significant findings. Full tables can go to an appendix in my opinion. What is the number in parentheses? I feel they should be significantly simplified.

Response: Thank you for your feedback regarding Tables III, IV, and V. We appreciate your suggestion to simplify the presentation of the findings. However, we feel these tables are central to our manuscript as they present the main results of our analysis. To address this, we propose to enhance the captions of these tables to clarify their contents and the meaning of the numbers in parentheses and help the reader interpret them. We also ensure that the discussion of these tables in the manuscript text clearly identifies and elaborates on the most significant findings to guide readers more effectively.

Please let us know if you still feel this needs further revision.

Link to change in the paper: Improved caption on table III only (to not repeat)

13. Dataset Public Availability

Comment: Is the dataset publicly available?

Response: Same response as Reviewer 1 above.

Thank you for your suggestion regarding the possibility of making the data openly accessible. We appreciate the potential value that such data could offer to the broader research community. However, the data cannot be distributed because it contains PII (personally identifiable information) in the form of GPS data. Many of these drivers are GM employees driving the cars to and from their homes. The authors are open to further conversation on this topic if it would be helpful.

Link to change in the paper: We have added a data availability statement at the end of the paper.

REVIEWERS' COMMENTS (STAGE 2)

Reviewer #1 (Remarks to the Author):

The reviewer's comments have been mostly addressed. The manuscript is almost ready for publication. Two minor comments:

A) On the rebuttal, the authors mention on point 7 that "The tested ACC systems track both the first and second vehicle ahead as separate objects. The position, velocity, and acceleration of these vehicles are used as separate, distinct inputs in the ACC system's decision-making process. The system therefore has potentially the capability to react to events like a sudden slowdown occurring several vehicles ahead, even if the vehicle directly in front has not yet reacted. However, while ACC systems have this capability, their current effectiveness in utilizing this data to employ strategies that optimize fuel efficiency is not clear". The reviewer believes that this may be useful information, that could be added in the discussion.

Response: This additional detail has been included in the manuscript, in the section about multi-anticipation in "Future Research Directions."

B) In the previous notes, I failed to explain what is the Regulation 79 I referred to. It is not EU Regulation 79, but UN Regulation No. 79 (<https://unece.org/transport/documents/2023/10/working-documents/un-regulation-no-79-revision-5>). I apologize for that.

Response: Thank you for this clarification. References have been accordingly updated in the manuscript.

Reviewer #2 (Remarks to the Author):

I would like to thank the authors for carefully replying to my comments and remarks. I have only a couple of final comments and I think that the paper can be accepted for publication.

Authors' reply to Comment 7: One minor comment is that statements like the following can be misleading as they present facts (or insights) without discussing the way these insights were investigated. I would therefore suggest removing this discussion or put it in the conclusion as preliminary findings for future research:

"Further analysis (not presented here) revealed that most trips are high functional class driving, typically consisting of local, short journeys with occasional highway usage."

Response: Thank you for this comment. Instead of eliminating this discussion, we provided more detail about exactly how the road attributes were determined using the available data.

Authors' reply to Comment 8: I would argue that this comment is not fully tackled. Although I agree with the authors' argumentation below, I still believe that it is important to state whether the main findings are still observed when the dataset is randomly reduced by e.g. 20%. Of course, I understand that the error will increase with a smaller dataset, so no absolute numbers are necessary. Nevertheless, the main key findings about the energy behavior of ACC systems should be there for a slightly reduced number of observations.

“However, we can provide some general insights based on statistical principles. More data generally leads to better results in statistical analysis. As the sample size (N) increases, the standard error decreases, leading to more precise estimates and reduced uncertainty.”

Response: We appreciate this recommendation and have incorporated it into the main body of the text, where we suggest random reduction of the existing dataset as part of future research investigating the sufficiency of this data. Passage from the revised manuscript: “Future research could include a well-crafted data sufficiency study to determine the minimum data requirements for robust ATE analysis. This study could include randomly reducing the existing dataset (e.g., by 20 percent) to test if the main findings are still uniformly observed in all reduced datasets.”